# Long-Horizon Planning for Multi-Agent Robots in Partially Observable Environments

**Siddharth Nayak**[1*]   **Adelmo Morrison Orozco**[1*]   **Marina Ten Have**[1]   **Vittal Thirumalai**[1]
**Jackson Zhang**[1]   **Darren Chen**[1]   **Aditya Kapoor**[2]   **Eric Robinson**[3]   **Karthik Gopalakrishnan**[4]
**James Harrison**[5]   **Brian Ichter**[† 5]   **Anuj Mahajan**[‡ 6]   **Hamsa Balakrishnan**[1]
[1]MIT   [2]TCS   [3]USAF-MIT AI Accelerator
[4]Stanford   [5]Google DeepMind   [6]Apple

## Abstract

The ability of Language Models (LMs) to understand natural language makes them a powerful tool for parsing human instructions into task plans for autonomous robots. Unlike traditional planning methods that rely on domain-specific knowledge and handcrafted rules, LMs generalize from diverse data and adapt to various tasks with minimal tuning, acting as a compressed knowledge base. However, LMs in their standard form face challenges with long-horizon tasks, particularly in partially observable multi-agent settings. We propose an LM-based Long-Horizon Planner for Multi-Agent Robotics (LLaMAR), a cognitive architecture for planning that achieves state-of-the-art results in long-horizon tasks within partially observable environments. LLaMAR employs a plan-act-correct-verify framework, allowing self-correction from action execution feedback without relying on oracles or simulators. Additionally, we present MAP-THOR, a comprehensive test suite encompassing household tasks of varying complexity within the AI2-THOR environment. Experiments show that LLaMAR achieves a 30% higher success rate than other state-of-the-art LM-based multi-agent planners in MAP-THOR and Search & Rescue tasks. Code can be found at https://github.com/nsidn98/LLaMAR

## 1   Introduction

Creating embodied agents that assist humans in real-life scenarios is a significant challenge when humans communicate their intentions in natural language. Certain tasks like moving furniture [1, 2], search-and-rescue [3], environmental monitoring [4], etc,. require coordination among multiple agents to solve the tasks efficiently as compared to single-agent scenarios. This challenge of understanding these natural language inputs and effectively coordinating these agents to solve the task is exacerbated in the multi-agent scenario. Recent works [5, 6, 7, 8, 9, 10, 11, 12, 13] have shown that Language Models (LMs) [1] can effectively use language instructions to develop plans for robots. However, most studies focus on single-agent long-horizon task planning. Naïve extensions of single-agent planning algorithms to multi-agent settings often fail due to environment non-stationarity, where the policies of other agents—modeled as a part of the environment—are continuously changing [14, 15]. Such failures lead to suboptimal performance as agents struggle to anticipate and adapt to the actions of others. We therefore formulate a centralized process in which decisions are made simultaneously for all agents based on their (partial) observations, similar to the centralized multi-agent system framework (CMAS) proposed in [16]. Leveraging the ability of pre-trained LMs to generalize across diverse tasks, we aim to use LMs for long-horizon embodied multi-agent task planning.

The key insight of our work is that integrating a *plan-act-correct-verify* framework with LMs enables a robust and adaptive approach to multi-agent task planning in dynamic, partially observable

---

[*]   Equal Contribution.   [†]   Now at Physical Intelligence.   [‡]   Work done outside Apple.   [1]   We denote Large Language Models as LLMs, Vision Language Models as VLMs

38th Conference on Neural Information Processing Systems (NeurIPS 2024).

environments that allows agents to: (1) plan subtasks required to complete the task, (2) select high-level actions for each agent to complete the proposed subtasks, (3) identify and correct failures after high-level action execution, and (4) self-verify subtask completion based on high-level action execution. Unlike existing methods, our approach uses real-time execution feedback, observations, and agent histories to iteratively refine action planning and execution. This allows agents to adjust strategies based on reasoned insights on action execution, effectively addressing failures without relying on perfect environmental knowledge or oracle feedback. The correction and verification process in our cognitive architecture [17] is grounded in the environment's reality, which sets it apart from LM self-verification methods that lack such grounding [18]. This framework enhances agents' ability to complete complex, long-horizon tasks, yielding substantial improvement over current state-of-the-art methods.

Similar to our approach, recent works [19, 20, 21, 22, 23, 24, 16] utilize LMs for multi-agent planning, often adopting a hierarchical decision-making structure. The LMs are used for high-level planning to determine subtasks, sometimes in conjunction with planning domain definition language (PDDL) that together with the LM planner, functions as a feasibility solver. Specific actions are executed using low-level policies pre-trained through reinforcement learning, behavior cloning, or heuristic approaches. While these methods effectively use LMs as high-level planners, they assume perfect low-level primitive action policies and simulator or oracle-provided environmental information. By contrast, LLaMAR does not assume perfect knowledge of the environment, does not rely on oracle feedback, and does not assume perfect execution of low-level primitive policies. This approach moves us closer to enabling real-world robots that operate independently of privileged knowledge.

To avoid ambiguity, we use the following conventions. We refer to the objectives within the environments as "tasks" or "goals" and "subtasks" to describe the breakdown of tasks or goals. "High-level actions" are defined as skills the agent can perform, while "low-level actions" or "primitive actions" refer to existing policies—either learned or predefined using heuristics—that execute a sequence of actions to accomplish a high-level action. More details and examples can be found in Appendix A.

The main contributions of this paper are:

- **LLaMAR**: An LM-based Long-Horizon Planner for Multi-Agent Robotics, designed for iterative planning of long-horizon, multi-objective tasks in partially observable environments, with the following key features:
  - It operates without prior knowledge of the environment, allowing agents to explore and make decisions based on new observations.
  - It evaluates outcomes through direct observation of images, rather than relying on oracles for feedback, enabling independent identification and correction of action failures.
- **MAP-THOR (Multi-Agent Planning in THOR)**: a benchmark suite of tasks within the AI2-THOR simulator under partial observability to standardize methodologies and metrics for evaluating multi-agent planning effectiveness and robustness.

## 2 Related Work

**Reinforcement Learning (RL) for Long-Horizon Planning**: While RL algorithms have shown promise in many applications, they still struggle with long-horizon tasks. Hierarchical reinforcement learning (HRL) has been used to address these challenges in both single-agent [25, 26, 27, 28] and multi-agent settings [29, 30, 31]. However, these approaches are typically applied to single-task, stationary environments, such as games, where agents solve for one goal in a fixed environment. Consequently, these methods do not generalize well across multiple environments or tasks. Multi-task RL has been explored as a potential solution, requiring sophisticated task planning to handle diverse objectives [32, 33]. This often involves decomposing tasks into manageable subtasks, a process well-suited for hierarchical frameworks. However, subtasks are known apriori in multi-task RL formulations. Real-world long-horizon RL necessitates robust task planning, and LMs have emerged as a promising approach for this purpose.

**LMs for Embodied Single-Agent Planning**: Recent studies have demonstrated the effectiveness of LMs in generating and executing plans in embodied single-agent environments [34, 35, 36, 37, 38, 39, 40] and creating plans in single-agent embodied robotic environments [41, 42, 43, 44, 45, 46, 47, 48, 49, 50, 51, 52]. Works like SayCan [5] and Grounded Decoding [6] use a combination of value functions and LLM predictions for long-horizon tasks. ProgPrompt [8] and Zero-Shot Language Planner [13] generate static plans executed in the environment, which may fail in partially observable

and dynamic settings. To mitigate this, LLM-planner [53] updates plans based on new observations, similar to our approach.

**LMs for Multi-Agent Planning**: Xu et al. [54] use LLMs in multi-agent games, while CoNavGPT [23] creates global plans for two robots in an embodied environment. RoCo [24] and CoELA [22] assign separate LMs to each agent for decentralized action prediction, allowing natural language communication between agents. However, RoCo and CoNavGPT require detailed environment information for planning, and CoELA's action space is filtered by an oracle. Relying on privileged information from an oracle is impractical in real-world applications. By contrast, our work focuses on free-form action generation and handles tasks with more ambiguous descriptions. Prior work [16] compare centralized (CMAS) and decentralized (DMAS) planning frameworks, showing that centralized planners perform better, though their experiments are in simple, known environments with limited number of agents. Two-Step [21] decomposes goals for main and helper agents, using PDDL planners for high-level actions. SmartLLM [19] uses multiple LLM modules for subtask decomposition, multi-robot group formation and task allocation but assumes robots have complete knowledge of the environment, making plans prone to errors in unknown settings. S-ATLAS [20] use LLMs with conformal prediction for safe multi-agent planning, but the action choices are limited to a small set of objects. Table 1 presents a comparison of the characteristics of different LM-based approaches to multi-agent planning with our work.

LMs can interpret high-level instructions and break them down into feasible subtasks, making them ideal for long-horizon, multi-task scenarios. Our work leverages LMs to enable long-horizon planning across a variety of tasks and environments, building on these advances to address the limitations of traditional RL and HRL methods. By integrating LMs into our planning framework, we enhance the ability to generalize across diverse tasks and scenarios, making significant strides toward practical, real-world applications of RL in dynamic, multi-agent settings.

| Method | Dynamic Planning | Local Information | Failure Correction | Self Verification |
|---|---|---|---|---|
| Two-Step [21] | ✗ | ✗ | ✗ | ✗ |
| Smart LLM [19] | ✗ | ✗ | ✗ | ✗ |
| S-ATLAS [20] | ✓ | ✗ | ✗ | ✗ |
| CoELA [22] | ✓ | ✓ | ✗ | ✗ |
| LLaMAR (this paper) | ✓ | ✓ | ✓ | ✓ |

Table 1: The proposed model, LLaMAR: 1) performs dynamic planning, avoiding the open-loop plan-and-execute paradigm; 2) operates without privileged simulator information (e.g., access to all objects in the environment); 3) re-plans when low-level actions fail, not assuming perfect execution; and 4) self-verifies subtask completion without relying on the simulator.

## 3 Background

**Problem Setting**: We consider a setting where multiple robots perform a series of tasks (a) such as cleaning a room or putting groceries in the fridge, in a home-like environment, and (b) rescuing missing personnel and putting out forest fires in a search & rescue environment (SAR). These tasks typically require long-horizon planning, involving around 100 low-level actions to reach the goal. Our objective is to compute plans for a team of robots to execute high-level language instructions, $I$. We formalize these tasks as partially observable Markov decision processes (POMDP) [55, 56], denoted as $\langle N, \mathcal{I}, \mathcal{S}, \{\mathcal{O}_i\}, \{\mathcal{A}_i\}, \mathcal{P}, \mathcal{G}, T \rangle$. $N$ is the number of agents and $\mathcal{I}$ is the high-level language instruction set. Here, $s \in \mathcal{S}$ represents the joint state of all agents, and $o \in \mathcal{O}$ denotes the observation set for all agents. Particularly, $o_i \in \mathcal{O}_i$ is the observation set of agent $i$, that captures incomplete environment state information. $a \in \mathcal{A} = \mathcal{A}_1 \times \mathcal{A}_2 \cdots \mathcal{A}_N$ represents the joint action space. The joint action space comprises of different categories of high-level actions $\mathcal{A} = \mathcal{A}_{NAV} \cup \mathcal{A}_{INT} \cup \mathcal{A}_{EXP}$, where $\mathcal{A}_{NAV}$ is the joint navigation action set, $\mathcal{A}_{INT}$ is the joint interaction actions which allow the agents to interact with objects, and $\mathcal{A}_{EXP}$ are the joint exploration actions which allow the agents to explore the environment. Examples of the high-level actions include `PickUp(object)` $\in \mathcal{A}_{INT}$ and `NavigateTo(location)` $\in \mathcal{A}_{NAV}$. Each high-level action is associated with a low-level primitive action (pre-trained RL, behavior cloned, or heuristic-based policy). These actions are executed *synchronously* by all agents at every high-level decision step. $\mathcal{P}(s'|s, a)$ is the joint transition probability function that defines the probability of arriving at $s' \in \mathcal{S}$ after taking joint

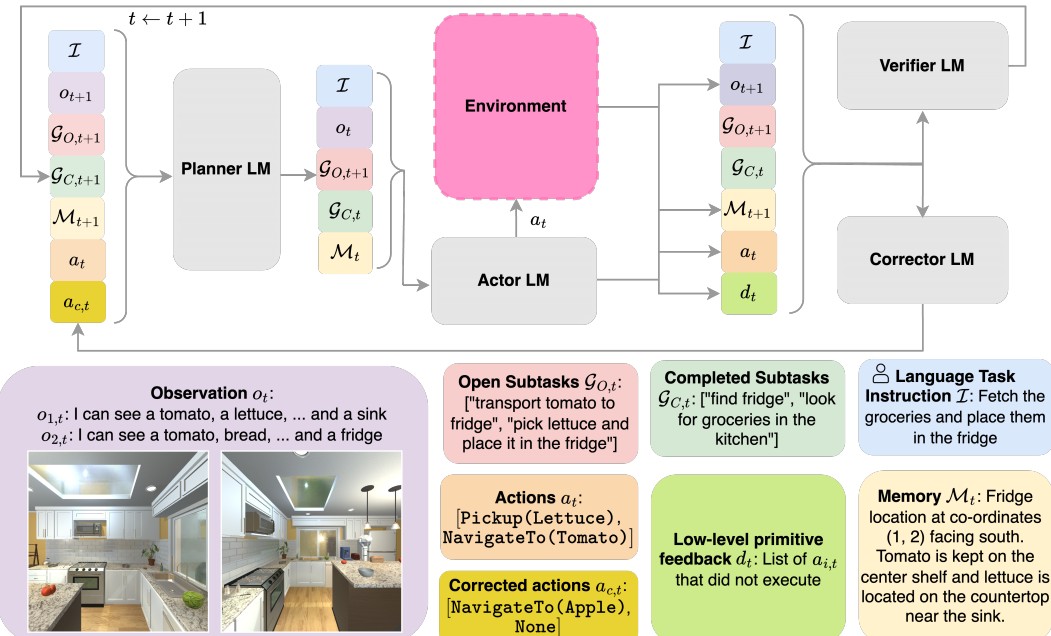

Figure 1: An overview of LLaMAR's modular cognitive architecture. LLaMAR leverages LMs within four key modules: Planner, Actor, Corrector, and Verifier, each with specific roles. The Planner breaks down the high-level language instruction into feasible subtasks to achieve the environment goal. The Actor determines the high-level actions each agent should perform. These actions trigger low-level policies that generate and execute a sequence of primitive actions in sync across all agents. Based on execution feedback, the Corrector suggests corrections for high-level actions and the Verifier Module validates completion of subtasks.

action $a \in \mathcal{A}$ in $s \in \mathcal{S}$. $\mathcal{G} = \{g_1, \cdots, g_k\}$ defines the subtasks that the agents need to perform to accomplish the language instruction task. $T$ is the length of the planning horizon.

**Environment**: To simulate open-ended, long-horizon tasks that resemble everyday activities in a home-like environment, we use the AI2Thor simulator [41], which supports a diverse set of interactions and photorealistic rendering. Since our approach does not require any parametric training, it can potentially translate to other similar embodied environments like VirtualHome [57], Habitat [42, 58, 59], and ThreeDWorld [60], possibly extending beyond household domains. Similarly, to simulate the search & rescue scenario, we create a custom Search & Rescue environment (SAR). More information about the MAP-THOR and SAR environments can be found in Appendix B and D respectively. We instantiate the problem with $N$ agents cooperating to accomplish a long-horizon rearrangement task [61] in an indoor environment. The agents do not know the objects present in the environment *a priori* and are encouraged to explore the environment to gather more information to complete the task. Unlike previous solutions in similar settings [22, 19, 20], we do not rely on an oracle/simulator to verify subtask completion. In prior works, a predefined conditional satisfiability check is used as subtask completion feedback.

## 4 Approach

We describe our approach in this section. Figure 1 illustrates LLaMAR's architecture comprising four modules: *Planner*, *Actor*, *Corrector*, and *Verifier*, each an LM with a distinct role. Prior work [62] shows that splitting roles across different LMs improves performance in sequential decision-making. Our initial experiments confirm that LMs tasked with reasoning about multiple inputs and providing long outputs perform poorly. We iterate through these four modules at every high-level decision step. The pseudocode for our approach is in Appendix E. We define some key notation below:

- **Memory** $\mathcal{M}$: A textual description of the joint memory of all agents, summarizing past observations, high-level actions, plausible reasons for action failures, and specific subtasks that each agent is attempting to solve.
- **Open Subtasks** $\mathcal{G}_O \subset \mathcal{G}$: Feasible subtasks proposed by the *Planner* LM to achieve the environment task that are yet to be accomplished by the agents.

- **Completed Subtasks** $\mathcal{G}_S \subset \mathcal{G}$: Subtasks completed by the agents.
- **Corrective Actions** $a_c$: Corrective actions for each agent based on failure information from the previous step.

At the start of each episode, Memory $\mathcal{M}$, Open Subtasks $\mathcal{G}_O$, Completed Subtasks $\mathcal{G}_S$, Actions $a$, Corrective Actions $a_c$, and Failure Information $\mathcal{F}$ are initialized as empty sets.

Consider an example of a kitchen with groceries, a fridge, and a counter. Two agents are tasked with "Fetch the groceries and place them in the fridge". This example will help illustrate the utility of each module. All LMs receive a language task instruction $\mathcal{I}$, joint observations from all agents, and information about open and completed subtasks and memory unless stated otherwise. We next discuss the various components in our architecture in detail:

**Planner Module** The *Planner* LM module suggests feasible subtasks to ensure the completion of the environment task. This method, similar to SmartLLM's [19] zero-shot planning, uses only observations in the agents' field of view. The *Planner* suggests subtasks related to objects seen in the current observation or memory of all the agents. For the example considered, it decomposes the task into subtasks like "transport the tomato to the fridge" and "transport the lettuce to the fridge", which are added to $\mathcal{G}_O$.

**Actor Module** The *Actor* LM additionally uses corrective actions suggested by the *Corrector* module in the previous time step to predict high-level actions for the current step. These actions are then executed in the environment to progress a subset of subtasks in the open subtask set $\mathcal{G}_O$ and accordingly updates the joint memory. For instance, the *Actor* module might suggest actions such as $a = [\texttt{Pickup(Tomato)}, \texttt{NavigateTo(Lettuce)}]$, updating memory with "We saw a tomato on the counter-top, Alice is picking up the tomato, and Bob is navigating to the lettuce".

**Corrector Module** The *Corrector* LM self-corrects high-level actions suggested by the *Actor* LM after controller failures in the previous step's execution[2]. It suggests corrective high-level actions and provides reasons for failures and chosen corrections. For example, it might suggest "The action of picking up the tomato failed because it is too far away. Alice first needs to navigate closer to the tomato."; $a_c = [\texttt{NavigateTo(Tomato)}, \texttt{None}]$.

**Verifier Module** After executing high-level actions, the *Verifier* LM assesses whether these actions have completed any subtasks in the open subtask set. Successful subtasks are moved to the completed subtask set. Without the *Verifier* LM, the method would need to rely on the simulator/oracle for success or failure information. The *Verifier* LM along with other information uses the successfully executed high-level actions proposed by the *Actor* LM to predict subtask completion. For example, after transporting the lettuce to the fridge, the *Verifier* updates the completed subtasks with "transport lettuce to the fridge".

**Admissible Action parsing with Semantic Translation** When LMs generate action plans, natural language outputs often fail to translate to executable high-level actions. This happens when the output does not match the predefined format or refers to unrecognized contextually similar objects. We use a cosine similarity method from [13], fine-tuning a pre-trained sentence-BERT [63] to transform the free-form text into admissible high-level actions. Hyperparameters and additional details of the sentence transformer fine-tuning are provided in Appendix J.

**Exploration Strategy** In unexplored environments, agents need to search for task-relevant objects. If agents cannot find the required objects, the language model can choose an 'exploration' action $a_{exp}$. We use a semantically-guided heuristic to determine the choice of region to be explored. The agent rotates to four cardinal directions $d \in North, South, East, West$, capturing image observations $o_{n,d}$. These images are processed through a pre-trained CLIP image encoder [64] to obtain embeddings $I_d$. The list of open subtasks $\mathcal{G}_O$ is processed through the corresponding CLIP text encoder to get text embeddings $g_{O,i}$. The exploration score $\mathcal{E}_d$ in direction $d$ is defined as $\mathcal{E}_d = \sum_{i=1}^{|\mathcal{G}_O|} \frac{g_{O,i} \cdot I_d}{\|g_{O,i}\| \|I_d\|}$. The direction with the highest score $d^* = \arg\max_d \mathcal{E}_d$ is chosen. Summing the scores helps select the best direction to explore in expectation. The agent rotates towards $d^*$ and moves $J = 2$ steps, repeating this process $K = 3$ times in one *explore* action. This approach ensures that images relevant to identifying potential subtasks are prioritized. For example, if $\mathcal{G}_O$ includes "locate a computer", it is more likely to find a computer on a table than on a sofa, resulting in a higher

---

[2] We use the simulator just to provide a boolean value about the success of high-level action execution.

cosine similarity score between the subtask CLIP text embedding and table CLIP image embedding. Refer to Appendix B.2 for more details about the exploration heuristic.

**Motivation for Proposed Framework** The specific order of the modules in LLaMAR is due to natural causal relationships in which environment feedback is received. We use the *Planner* as the first module because it allows LLaMAR to come up with an initial list of open subtasks that could be completed based on the current observation and past memory to satisfy the task. This list serves as a rough high-level plan. The actor then uses this information to suggest the necessary actions. The *Corrector* is used after the *Actor* module to identify reasons for failures in the execution of the actions suggested by the *Actor* . Note that the failure module is inert and only suggests corrective actions. Only the *Actor* module decides the final actions to be executed. This role distinction allows for clear reasoning on failures when they occur and lets the actor module focus on choosing actions. The *Verifier* is used after the action is executed to update the list of closed subtasks so that LLaMAR can be current with the progress toward the completion of the environment task. This allows the planner to update the list of open subtasks in the next step. In essence, the *Planner* and the *Verifier* ensure that the progress of the agents is tracked and the actor and the corrector ensure that the actions are executed successfully to advance towards completion of the task.

**Multi-Agent features in LLaMAR** While our method can be easily adapted to a single-agent setting, our design choice for the architecture was motivated to include the following multi-agent features:

- **Coordination through communication**: Agents share their state information with the centralized LLaMAR modules to predict actions, enabling them to coordinate and avoid conflicts. This information sharing allows for the agents to cooperate and achieve the collective goal.
- **Dynamic Role Assignment**: Agents are dynamically assigned roles based on the current task requirements and their capabilities. This flexibility allows LLaMAR to adapt to changing environments and task demands.
- **Hierarchical Task Decomposition**: To handle the complexity of multi-agent planning, LLaMAR decomposes the action space by creating specific subgoals/subtasks available for any agent to assign itself (done by the actor module) based on the observation and current context. This decomposition reduces the overall search space and improves planning efficiency.

## 5 Experiments

**MAP-THOR**: To evaluate the performance of LLaMAR and benchmark other baseline methods, we create a benchmark dataset of tasks which we call MAP-THOR (Multi-Agent Planning tasks in AI2-THOR). While Smart-LLM [19] introduces a dataset of 36 tasks within AI2-Thor [41] classified by complexity, their tasks are limited to single floor plans. This limitation hinders testing the robustness of planners across different room layouts. Additionally, some tasks in their dataset cannot be performed by multiple agents, regardless of task division, such as `Pick up the pillow`, `Open the laptop to turn it on`, and `Turn off the lamp`.

By contrast, MAP-THOR includes tasks solvable by both single and multiple agents. We classify the tasks into four categories based on the ambiguity of the language instructions. To test the planner robustness, we provide five different floor plans for each task. We also include automatic checker modules to verify subtask completion and evaluate plan quality. Our dataset comprises 45 tasks, each defined for five distinct floor plans, ensuring comprehensive testing and evaluation.

We conduct experiments with tasks of varying difficulty levels, where an increase in difficulty of the tasks corresponds to an increased ambiguity in the language instructions. The complete task list of each category can be found in the Appendix C.

- **Explicit item type, quantity, and target location**: Agents are explicitly instructed to transport specific items to specific target locations. For example, `put bread, lettuce, and a tomato in the fridge` clearly defines the objects (tomato, lettuce, bread) and the target (fridge).
- **Explicit item type and target location but implicit item quantity**: The object type is explicitly described, but its quantity is not disclosed. For example, `Put all the apples in the fridge`. Agents must explore the environment to locate all specified items and also predict when to stop.
- **Explicit target location but implicit item types and quantity**: The target location is explicitly defined but the item types and their quantities are concealed. For example, `Put all groceries in the fridge`.

- **Implicit target location, item type, and quantity**: Item types and their quantities along with the target location are implicitly defined. For example, `Clear the floor by placing the items at their appropriate positions.` The agent is expected to place items like pens, books, and laptops on the study table, and litter in the trash can.

**Search & Rescue Environment (SAR)**: To showcase the effectiveness of LLaMAR with respect to explicit coordination in multi-agent settings, we evaluate LLaMAR in a partially observable search & rescue and fire relief environment in a grid world. Depending on the scene, there is a mix of missing people to be found, and wildfires to be stopped before they spread geographically. More details about the environment can be found in Appendix D.

- **Fire Extinguishing**: Fires consist of expansive flammable regions with a fixed set of sources that propagate over time. The rate of fire spread is proportional to its intensity; higher intensities result in faster spread. Fires are categorized as either Class A or Class B, which are extinguished using water or sand, respectively. These extinguishing resources are sourced from reservoirs distributed across the environment.
- **Human Rescue**: Each individual is initially located at an unknown position within the environment. The objective is to locate, rescue, and transport the individuals to a designated drop-off location, which is known beforehand. Transporting a person requires the coordinated effort of two agents simultaneously, who must carry them to the specified drop-off point.

### Metrics

We evaluate the algorithms using the following metrics to compare their performances on the tasks:

- **Success Rate** (SR): The fraction of episodes in which all subtasks are completed. Success equals 1 if all subtasks are successfully executed in an episode, otherwise it is 0.
- **Transport Rate** (TR): The fraction of subtasks completed within an episode, provides a finer granularity of task completion.
- **Coverage** (C): The fraction of successful interactions with target objects. It is useful to verify if the LMs can infer the objects to interact with, in scenarios where the tasks have objects that are specified implicitly.
- **Balance** (B): The ratio between the minimum and maximum number of successful high-level actions executed by any agent that contributed towards task completion. We only check for a subset of high-level actions that must be executed to accomplish critical subtasks that lead to the successful completion of the language instruction task. If each agent $i$ out of $n$ agents completes $s_i$ successful tasks, the balance is defined as: $B := \frac{\min\{s_1, \cdots, s_n\}}{\max\{s_1, \cdots, s_n\} + \epsilon}$. This measures how evenly the work is distributed among agents. A balance of zero indicates at least one agent performed no successful high-level actions, while a balance of one indicates all agents performed the same number of successful high-level actions. Here $\epsilon = 1e - 4$ is a small number to avoid division by zero.
- **Average steps** (L): The number of high-level actions taken by the team to complete the task, capped at $L = 30$ in our experiments. If the task is not completed within $L$ steps, the episode is deemed a failure. Note that the metric $L$ is presented in the table providing the complete results, located in Appendix F.

For all the metrics, we report the means along with the $95\%$ confidence interval across all the tasks (refer Appendix F for complete results). Since SR is a binomial metric, we report the Clopper-Pearson Interval as the confidence interval.

### Baselines

For a fair comparison with our method, we make modifications to the baselines to make them work in partially observable settings with limited reliance on the simulator. More details about implementations can be found in Appendix H.

- **Act**: We query the LLM with the task and the observations to suggest a high-level action.
- **Chain-of-Thought** [65]: We modify the Act prompt with a chain-of-thought style addendum to let the LM reason about the possible implications while selecting a high-level action.
- **ReAct** [66]: We use a ReAct-style prompting to let the LMs reason after suggesting high-level actions and possibly suggest ways to correct any failures.
- **SmartLLM** [19]: We modify the official codebase to only include information from the local observations of the agents instead of assuming full observability.

- **CoELA** [22]: We modify the list of available high-level actions to include all possible valid combinations of actions with interactable objects in the agent's local observation. As the scene becomes more cluttered, this list and the prompt becomes combinatorially longer. In the original implementation, the list of available actions is filtered based on the feasibility of the actions as suggested by a conditional checker.

It should be noted that Act, Chain-of-Thought, ReAct, and SmartLLM are all CMAS frameworks where CoELA follows the DMAS framework.

## 6    Results and Discussion

**Choice of the underlying LM**: To understand the impact of the underlying LM's quality on decision-making, we initially experimented with different LMs on MAP-THOR. Specifically, we utilize both the language-only and vision-language models of GPT-4 [67], IDEFICS-2 [68, 69], LLaVA [70, 71], and CoGVLM [72]. Among these, GPT-4, when used solely with text inputs, exhibits the poorest performance. This is attributed to the agents' inability to reason about visual observations, which is particularly detrimental for the *Corrector* module. Substituting GPT-4V with other vision-language models results in a decline in performance (refer Table 2) and hence we use GPT-4V as the underlying VLM while comparing to the baselines.

**Baseline Comparisons**: Table 2 compares our method, LLaMAR, with other baselines in a 2-agent scenario using GPT-4V as the underlying VLM. Act and ReAct show similar performance, with Act struggling due to its lack of strategic planning or correction, and ReAct performing slightly better by dynamically adjusting actions based on reasoning on immediate feedback. CoT's performance declines with longer planning horizons due to its inability to maintain coherence over extended planning sequences, consistent with findings in [73], showing its effectiveness only with highly specific prompts. SmartLLM, operating in a *plan-and-execute* paradigm, generates impractical plans with issues like infinite loops and failure to handle low-level action failures, leading to lower success rates and poor transport metrics. It also tends to hallucinate objects. CoELA, using a decentralized multi-agent system (DMAS), performs poorly due to large input prompts and struggles to select the correct action from numerous choices. Its decentralized decision-making is less efficient than the centralized multi-agent system (CMAS) used by LLaMAR. Previous research [16] confirms CMAS frameworks are more effective than DMAS frameworks. Overall, our method, LLaMAR, benefits from its modular cognitive architecture, which integrates planning, acting, correcting, and verifying through distinct LLM roles, resulting in superior performance across various evaluation metrics. By avoiding reliance on privileged information and incorporating a robust exploration strategy that allows it to scout for objects that are not initially visible, LLaMAR ensures higher success rates and balanced task execution among agents.

**Roles of different modules in LLaMAR**: To demonstrate the effectiveness of the various modules in our cognitive architecture, we performed ablation studies by evaluating performance metrics with each module removed individually. The results are summarized in Table 3. Using only the *Actor* module corresponds to the "Act" baseline, which demonstrates its fundamental capabilities in isolation but shows limited effectiveness without planning and correction due to relatively lower success and transport rates. Adding the Planner and Verifier modules improves performance, benefiting from better task planning and validation, increasing the overall SR and TR, and ensuring more effective task completion and even work distribution, as indicated by the increase in balance (B). However, in scenarios where the suggested action fails, the actor suggests the same action in the next decision step since it is not able to reason why the action failed until the end of the planning horizon. Incorporating the Corrector module, with access to privileged information from an environment oracle, significantly boosts performance, enhancing the SR, TR, C, and further improving B, consistent with the findings in [17]. This highlights the Corrector module's importance in adjusting actions based on controller feedback, resulting in higher task success and more efficient task completion, albeit with reliance on oracle knowledge. Finally, the complete LLaMAR system, without privileged information, achieves SR, TR, C, and B values close to those of the oracle setup. This demonstrates the system's robustness and effectiveness in a realistic setting. The Corrector module plays a crucial role in enabling agents to learn from past failures and avoid repeating actions, preventing task failures due to timeout. Despite lacking oracle knowledge, LLaMAR performs nearly as well as the oracle-enhanced setup. These results highlight the importance of each module in our cognitive architecture. Removing any module diminishes effectiveness.

**Increasing the number of agents** Increasing the number of agents in the environment shows distinct

| Algorithm | LM | SR↑ | TR↑ | C↑ | B↑ |
|---|---|---|---|---|---|
| Act | GPT-4V | 0.33 | 0.67 | 0.91 | 0.59 |
| ReAct | GPT-4V | 0.34 | 0.72 | 0.92 | 0.67 |
| CoT | GPT-4V | 0.14 | 0.59 | 0.87 | 0.62 |
| SmartLLM | GPT-4V | 0.11 | 0.23 | 0.91 | 0.45 |
| CoELA | GPT-4V | 0.25 | 0.46 | 0.76 | 0.73 |
| LLaMAR | GPT-4 | 0.51 | 0.85 | 0.95 | 0.83 |
| LLaMAR | LLaVA | 0.54 | 0.84 | 0.91 | 0.75 |
| LLaMAR | IDEFICS-2 | 0.57 | 0.86 | 0.94 | 0.78 |
| LLaMAR | CogVLM | 0.61 | 0.89 | 0.95 | 0.80 |
| LLaMAR (w/o expl) | GPT-4V | 0.62 | 0.87 | 0.95 | 0.82 |
| LLaMAR (w/ expl) | GPT-4V | **0.66** | **0.91** | **0.97** | **0.82** |

Table 2: Comparison of evaluation metrics against baselines averaged across all tasks for the 2-agent MAP-THOR scenarios. The complete table with confidence intervals can be found in Appendix F. More details about peculiar behaviors for the baselines can be found in Appendix H.

| Modules Used | SR ↑ | TR ↑ | C ↑ | B ↑ |
|---|---|---|---|---|
| Actor | 0.33 | 0.67 | 0.91 | 0.59 |
| Planner + Actor + Verifier | 0.45 | 0.78 | 0.92 | 0.69 |
| Planner + Actor + Corrector‡ | **0.67** | **0.91** | **0.97** | **0.84** |
| Planner + Actor + Corrector + Verifier + | 0.66 | **0.91** | **0.97** | 0.82 |

Table 3: Performance in the 2-agent scenarios in MAP-THOR obtained by ablating different modules in LLaMAR with GPT-4V as the underlying LM

| # of agents | MAP-THOR | | | | SAR | | | |
|---|---|---|---|---|---|---|---|---|
| | SR↑ | TR↑ | C↑ | B↑ | SR↑ | TR↑ | C↑ | B↑ |
| 1 | 0.37 | 0.67 | 0.87 | **1.00** | 0.28 | 0.75 | 0.86 | **1.00** |
| 2 | 0.62 | 0.87 | 0.95 | 0.82 | 0.44 | 0.86 | 0.94 | 0.91 |
| 3 | **0.70** | **0.91** | 0.98 | 0.66 | 0.68 | 0.92 | 0.96 | 0.80 |
| 4 | 0.68 | 0.90 | **0.99** | 0.62 | 0.72 | 0.94 | 0.98 | 0.78 |
| 5 | 0.62 | 0.90 | **0.99** | 0.54 | **0.74** | **0.96** | **1.00** | 0.73 |

Table 4: LLaMAR with various number of agents in the scenario in both MAP-THOR and SAR environments

trends in our method's performance metrics for both MAP-THOR and SAR environments (refer Table 4). In MAP-THOR, with two agents, we establish a solid baseline for success rate (SR) and transport rate (TR), which is similarly reflected in the SAR environment. Adding a third agent improves both SR and TR in both environments, indicating enhanced task completion and transportation efficiency. Coverage (C) also increases, suggesting better exploration and interaction with objects across both environments. There is a slight decrease in SR and TR when the number of agents increases from 3 to 5 in the MAP-THOR environment. The decrease in these metrics can be attributed to the rooms in MAP-THOR becoming crowded with 4 and 5 agents hence blocking the agents from navigating without colliding with other agents. But this phenomenon is not seen in the SAR environment which is comparatively more spacious and navigable. However, balance (B), which measures the even distribution of tasks among agents, decreases with more agents. This drop highlights the challenge of ensuring equal contributions from all agents in a larger multi-agent system. While SR remains high, the balance metric drops significantly from 2 to 5 agents, indicating some agents do more work than others. In summary, adding more agents improves task performance and efficiency but introduces challenges in maintaining balanced contributions. Addressing this imbalance is crucial for refining multi-agent planning algorithms.

**Correcting Failures**: In numerous instances, the actions proposed by the *Actor* module, such as `pick up <object>`, are unsuccessful due to the agent's insufficient proximity to the target object. In such situations, the *Corrector* module uses visual feedback to learn from these failures and recommends appropriate corrective actions, such as `navigate to <object>` to facilitate closer proximity. Figure 2 shows examples where the *Corrector* module interprets low-level action failures and suggests remedies, highlighting its importance.

# 7   Limitations and Future Work

**Higher number queries to the LM**: Since each high-level decision step requires querying 4 different LM-based modules, the cost and the compute times are higher than other baselines, especially compared to the plan-and-execute baselines like SmartLLM. An interesting future direction to

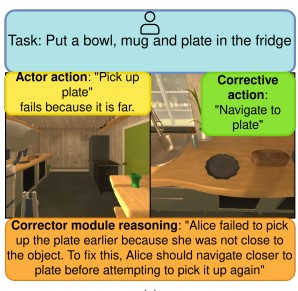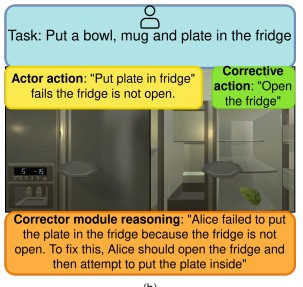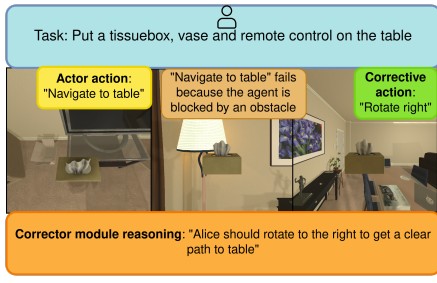

Figure 2: A few examples of the *Corrector* module mitigate failures in predicted actions by the *Actor* module. (a) the Corrector suggests getting closer to the agent before attempting to pick it up, (b) the Corrector recommends opening the fridge because the previous action of placing the plate failed, (c) the Corrector advises rotating right so that it can access the table to place the tissue box on it when the low-level navigation policy failed to find a path to the table

improve this would be to fine-tune smaller LMs with trajectories collected in the simulator (eg: ALFRED [46]) as done in [22]. Another potential direction worth exploring is using different sizes of LMs for each module based on their specific utility.

**Limited spatial reasoning**: Although we use both textual descriptions and visual features to guide the language model's actions, it still lacks the ability to reason about the spatial features of the environment. Spatial reasoning is crucial in scenarios such as navigating around obstacles to reach an object, or determining the shortest path to collect multiple items scattered across different locations. One way to address this limitation is to inject information about the 3D world into the LM, as done in [74], which is an interesting direction for future work.

**Performance limited by the underlying VLM**: Although LMs make correct reasoning most of the time, they still occasionally make mistakes, including misunderstanding the environment rules specified in the prompt. For example, the agent assumes that the cleaning task requires putting soap, drying, and putting it in the sink when all it needs is the action "*CleanObject*", and can't figure out the appropriate level of abstraction. The performance of the algorithm is limited by the instruction following and reasoning capability of the underlying LM [75, 76]; this calls for developing LMs that are fine-tuned to instruction-image pairs relevant to the environment (as done in [22]).

# 8 Conclusion

We address long-horizon planning in dynamic, partially observable multi-agent environments with LLaMAR, an LM-based planner using four specialized modules: *Planner*, *Actor*, *Corrector*, and *Verifier*. This framework iteratively refines action planning, adapts to failures, and verifies subtask completion using real-time observations and action feedback, without privileged information. We also introduce a heuristic-based exploration strategy to guide agents to semantically relevant regions. Additionally, we present MAP-THOR, a benchmark dataset for multi-agent tasks in the AI2Thor simulator. Empirical results show LLaMAR outperforms existing LM-based approaches, achieving a 30% higher success rate on MAP-THOR.

## Acknowledgements

We would like to thank Keerthana Gopalakrishnan, Sydney Dolan, Jasmine Aloor, and Victor Qin for helpful discussions and feedback. OpenAI credits for GPT-4 access was provided through OpenAI's Researcher Access Program. The research was sponsored by the Department of the Air Force Artificial Intelligence Accelerator and was accomplished under Cooperative Agreement Number FA8750-19-2-1000. The views and conclusions contained in this document are those of the authors and should not be interpreted as representing the official policies, either expressed or implied, of the Department of the Air Force or the U.S. Government. The U.S. Government is authorized to reproduce and distribute reprints for Government purposes notwithstanding any copyright notation herein.

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

# A  Terminology

We differentiate between the terms subtasks and high-level actions in this section. In essence, multiple high-level actions are needed to be carried out in a sequence to complete a subtask. Multiple subtasks need to be satisfied to complete the high-level language instruction.

- **Subtasks**: A task is split up into multiple subtasks. For example, if a task is "Fetch all the groceries and put them in the fridge", then the initial subtasks could include: "Locate the groceries", "transport the groceries", "Locate the fridge". These subtasks could get updated with new observations. For example, while locating the groceries, the agents come across a tomato and a lettuce. Then the subtasks "transport the tomato to the fridge" and "transport the lettuce to the fridge" gets updated in the subtasks list. This basically splits up the high-level instruction $\mathcal{I}$ into multiple mid-level subtasks
- **High-level actions**: These are the set of actions required to complete the subtasks. For example, to complete the "transport the lettuce in the fridge", we would require: the following set of actions:
  - Navigate to lettuce
  - Pickup lettuce
  - Navigate to the fridge
  - Open fridge
  - Put lettuce in the fridge
  - Close fridge

  Note that different agents have to complete different high-level actions that progress the subtasks efficiently whilst avoiding conflicts.
- Conflicts can arise in the following ways:
  - **Same actions**: Agents performing the same action at the same time. Example: "Open the fridge".
  - **Blocking**: Agent 1 is blocking Agent 2 and not allowing it to complete its high-level action. Example: Agent 1 is attempting to execute "PlaceObject(Tomato)" in front of the fridge to place the tomato in its hand in the fridge and Agent 2 is attempting to execute "OpenFreezer()" needs to interact with the fridge. Would require some form of conflict resolution in the state cell domain. Agent 1 should move away to allow fridge access to Agent 2. In LLaMAR, the *Corrector* module helps in figuring out these conflicts and suggest different corrective high-level actions.

# B  MAP-THOR Environment

The environment is based on the AI2Thor simulator with a multi-agent setup. All the experiments were performed in the single-room floor plans. When more than 3 agents are added to some of the floor plans (especially the kitchen floor plans), the environment gets crowded and does not allow for a lot of free space to navigate to different objects (the number of reachable paths reduces).

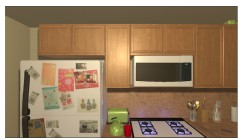 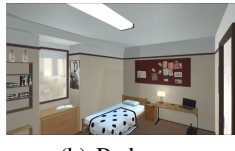 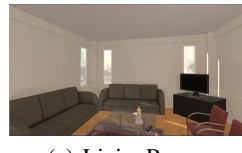 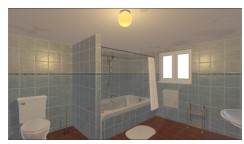

(a) Kitchen        (b) Bedroom        (c) LivingRoom        (d) Bathroom

Figure 3: Photorealistic rendering of household scenarios in the AI2Thor simulator enables the usage of multiple autonomous robots to carry out daily tasks.

## B.1  Observation Space

The observations for each robot include an image of size resolution $1000 \times 1000 \times 3$. The textual observation for each agent in the prompt is the list of objects visible in this image and the agents' current location and rotation. The field of view is 90 degrees. The agents can interact with the objects only if it is within its visibility range of $1.5m$.

## B.2 Action Space

The actions space $\mathcal{A}$ consists of navigation actions $\mathcal{A}_{NAV}$, interaction actions $\mathcal{A}_{INT}$, exploration action $\mathcal{A}_{EXP}$.

**Navigation actions** $\mathcal{A}_{NAV}$ consists of the following actions:

- `Move(<direction>)`: Moves the robot by $0.25$m towards the specified direction where `<direction>` can be one of (`Ahead, Back, Right, Left`)
- `Rotate(<direction>)`: Rotates the robot by $90$ degrees towards the specified direction where, `<direction>` can be one of (`Right, Left`)
- `LookUp(<angle>)` rotates the pitch of the robot camera upwards by the specified angle.
- `LookDown<angle>` rotates the pitch of the robot camera downwards by the specified angle.
- `NavigateTo(<object_id>)` makes the robot navigate to the specified object. The path is found using the $A^*-$shortest path algorithm. Note that the robot is only able to find a path to the specified object in the environment only if it has encountered that object previously during the episode. Otherwise, the `NavigateTo(.)` action will be unsuccessful and the agent will have to explore.

**Interaction actions** $\mathcal{A}_{INT}$ consists of the following actions:

- `Pickup(<object_id>)`: Picks up the object
- `Put(<receptacle_id>)`: Puts the object in the robots hand on the receptacle
- `Open(<object_id>)`: Opens the object
- `Close(<object_id>)`: Closes the open object
- `Slice(<object_id>)`: Slices the object
- `Clean(<object_id>)`: Cleans the object
- `ToggleOn(<object_id>)`: Toggles the object on
- `ToggleOff(<object_id>)`: Toggles the object off

**Explore action** $\mathcal{A}_{EXP}$: The explore action is carried out by the heuristic explained in Algorithm 1 and Figure 4. We use the `clip-vit-large-patch14-336` model for the CLIP weights which we download from https://huggingface.co/openai/clip-vit-large-patch14-336.

The action space consists of several executable actions, including `Move(<direction>)`, `Rotate(<direction>)`, `LookUp(<angle>)`, `LookDown(<angle>)`, `NavigateTo(<object_id>)`, `Pickup(<object_id>)`, `Put(<receptacle_id>)`, and others. These actions can be combined in numerous ways, as the `<object_id>`, `<directions>`, and `<angles>` can vary significantly, resulting in a combinatorially large action space. To address this complexity, we do not impose constraints on the language model (LM) modules to select actions in a predefined format from this set. Instead, we allow for free-form language outputs, offering greater expressivity. The SentenceBERT module then maps these free-form natural language outputs to executable actions within the environment.

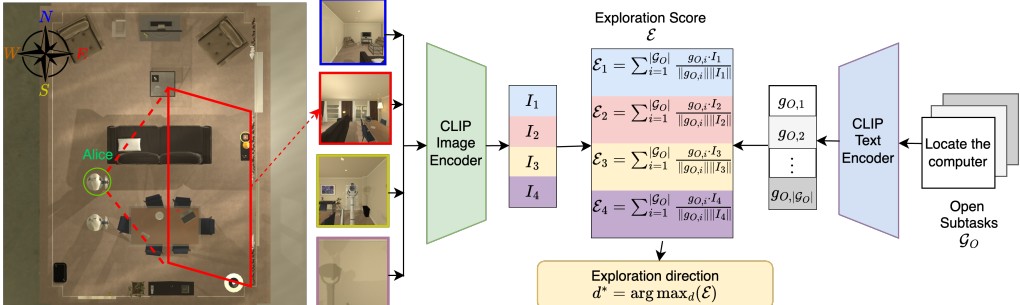

Figure 4: Choice of direction for the exploration heuristic: The agent (Alice) rotates towards 4 cardinal directions to get observations. The cosine similarity between the CLIP embeddings $I_d$ for these 4 images are calculated with the CLIP embeddings for each subtask in the open subtasks set $\mathcal{G}_O$ to get the exploration score $\mathcal{E}_d$ for each direction. The direction with the highest $\mathcal{E}_d$ is chosen to explore and the agent moves $J = 2$ steps in that direction.

**Algorithm 1** AI2Thor Exploration Heuristic

**Input**: Agent ID $n$, Environment $env$, number of exploration steps $K$, number of move steps $J$

1: $g_O = \text{CLIP}_{\text{text}}(\mathcal{G}_O)$
2: **while** $k < K$ **do**
3:     Exploration Score $\mathcal{E} \in \mathbb{R}^4 \leftarrow \mathbf{0}$
4:     **for** $d \in \{North, South, East, West\}$ **do**
5:         $o_{n,d} = env.step(\text{Rotate}(\text{Right}, \text{n}))$
6:         $I_d = \text{CLIP}_{\text{img}}(o_{n,d})$
7:         $\mathcal{E}_d = \frac{I_d \cdot g_O}{\|I_d\|\|g_O\|}$
8:     **end for**
9:     $d^* = \arg\max_d \mathcal{E}$
10:     **while** $j < J$ **do**
11:         $o_i = env.step(\text{Move}(d^*, n))$
12:         $j \leftarrow j + 1$
13:     **end while**
14:     $k \leftarrow k + 1$
15: **end while**

## C   MAP-THOR Task Types

The complete list of task for each task type:

- **Explicit object type, explicit quantity and target**:
  - put bread, lettuce, and a tomato in the fridge
  - Put the pots and pans on the stove burners
  - Slice the bread and tomato and crack the egg
  - Put the butter knife, bowl, and mug in the sink
  - Turn off the faucet and light if either is on
  - Put the tissue box, keys, and plate in the box
  - Put the computer, book, and pen on the couch
  - Put the bowl and tissue box on the table
  - Put apple in fridge and switch off the light
  - Put the watch and Keychain inside the drawer
  - Wash the bowl, mug, pot, and pan
  - Put the Box on the sofa and the bowl in the box
- **Explicit object type and explicit target**: Here we explicitly describe the object type but keep the quantity of the objects ambiguous. E.g. `Put all the apples in the fridge`. For this, the agents have to explore the environment to ensure that they find all of them.
  - Open all the drawers
  - Open all the cabinets
  - Turn on all the stove knobs
  - Put all the vases on the table
  - Put all the potatoes in the bowl
  - Put all pencils and pens in the box
  - Move all lamps next to the door
  - Turn off all light switches
  - Turn on all light switches
- **Explicit target, implicit object types**: The object types are implicitly defined whereas the target is explicitly defined. E.g. `Put all groceries in the fridge`. This tests whether the model can identify objects of certain categories.
  - Put all groceries in the fridge (should identify the tomato, bread, apple, potato, and lettuce)
  - Put all shakers in the closest drawer (should identify the salt shaker and pepper shaker)
  - Put all tableware on the countertop (should identify the bowl, plate, mug)
  - Put all food on the countertop (should identify the tomato, bread, apple, potato, and lettuce)
  - Put all school supplies on the couch (should identify the pencil, computer, and book)
  - Put all kitchenware in the cardboard box (should move the bowl and plate)
  - Put all silverware in the sink

- Move everything on the table to the desk (should move the laptop, pencil, pen, plate, credit card, book, and newspaper)
- Slice the lettuce, trash the mug and switch off the light
- Put all electronics on the couch
- Make a dish by microwaving eggs and tomato
- Put all readable objects on the sofa
- Wash all fruits

- **Implicit target and object types**: Here both the object type and the target are implicitly defined. E.g. `Clear the floor by placing the items at their appropriate positions`. Here the model is expected to keep items like pens, book, laptop on the study table, litter in the trash can, etc.
  - Clear the floor by placing items at their appropriate positions (depending on what's on the floor)
  - Clear the table by placing the items in their appropriate positions (depends on the floorplan, e.g. bread, apple, tomato, knife, bowl, book)
  - Clear the countertop by placing items in their appropriate positions (should move the lettuce, mug, and paper towel roll)
  - Clear the desk by placing the items in other appropriate positions (should move the statue, watch, and remote control)
  - Clear the table by placing the items in other appropriate positions (should move the book, credit card, laptop, plate, newspaper, pen, and pencil)
  - Clear the couch by placing the items in other appropriate positions (should move the pillow)
  - Make the living room dark
  - Make a mug of coffee and toast the bread
  - Trash all groceries
  - Slice all sliceable objects

# D   Search & Rescue Environment (SAR)

The Search & Rescue environment consists of multiple agents in an unknown environment that has multiple wildfires and missing personnel in the environment. The agents are tasked to extinguish all the fires before they spread and rescue the missing humans. Here, each fire is composed of a large flammable region with a fixed set of sources that spread through time. The higher the intensity, the faster the fire will spread. The fires can be of class A or B, extinguished through the use of water and sand respectively. Both these resources can be collected through resource reservoirs spread geographically. Each person is initially stranded in an unknown location. The goal is to rescue and transport each person to a drop-off location (known apriori). The person must have two agents simultaneously carrying them to be transported to the drop-off location.

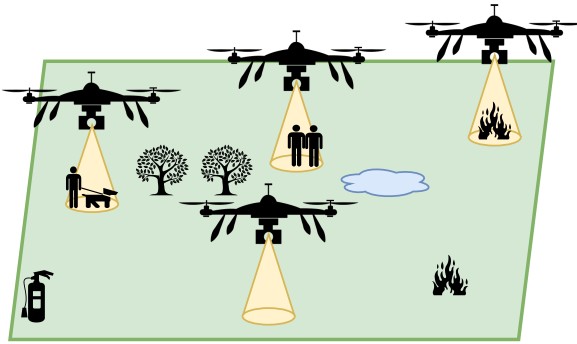

Figure 5: The search & rescue environment consists of multiple drones in an unknown environment with missing people, fires of different types, and water and sand reservoirs.

We provide a justification to why this task is relevant to long-term planning and multi-agent collaboration below.

### D.1 Multi-agent Nature

- **Scalability** - Unlike the AI2THOR environment, the search & rescue environment is more spacious and hence prevents congestion amongst agents. With the addition of multiple parallel emergency disaster scenarios at once, the environment scales comfortably to more agents. Additionally, the complexity slowly increases as the model has to coordinate between more actors and competing emergencies.
- **Tasks requires explicit cooperation** -
  - To move a group of humans, at least 2 agents are required. Thus, explicit multi-agent collaboration is required.
  - Due to the time pressure, in order to successfully stop the fire, agents must all effectively collaborate in collecting and strategically using resources in high-intensity regions.
- **Exploration & Task Assignment** - There is an explicit tradeoff between allocating agents towards exploring to find lost people and fighting the current fires.

### D.2 Long-term Planning

- **Dependency Chain**: To resolve a fire it requires that the source is identified, the type of fire is identified, and the appropriate resources to stop it are acquired and then used.
- **Uncertainty**: Inherent uncertainty as to where the lost people can be found, and at what point in the timeline they will.
- **Irreversibility and forced consequences** - With a fire that spreads, present actions have irreversible future consequences. A balance needs to be struck between fighting the fire's source (to stop it from continuing) versus a periphery (to prevent geographic spread).

### D.3 Description of Scenes

We evaluate LLaMAR in five different scenarios in the search & rescue environment. Here each scene is evaluated on 5 different random seeds.

- **Scene 1**: This consists of 1 Type A fire with 2 initial sources, 1 Type B fire with 1 initial source, and 1 lost person at a random location in the environment.
- **Scene 2**: This consists of 1 Type A fire with 1 initial source, 1 Type B fire with 2 initial sources, and 1 lost person at a random location in the environment.
- **Scene 3**: This consists of 2 Type A fires each with 1 initial source, 1 Type B fire with 1 initial source, and 1 lost person at a random location in the environment.
- **Scene 4**: This consists of 1 Type A fire with 3 initial sources, and 1 lost person at a random location in the environment.
- **Scene 5**: This consists of 1 Type A fire with 1 initial source, 1 Type B fire with 1 initial source, and 2 lost persons at random locations in the environment.

### D.4 Observation Space

The agent's observation $\mathcal{O} = \mathcal{O}_G \cup \mathcal{O}_L \cup \mathcal{O}_L$ is a union of the following observations.

**Global Observations** $\mathcal{O}_G$ consists of all the objects that are globally visible by the agent:

- `Fires`: If visible, fire name, type, and average intensity.
- `Fire Regions`: If fire visible, fire region name, type, and average intensity.
- `Reservoir`: If visible, reservoir name, and reservoir resource type.
- `Deposit`: If visible, deposit name, inventory of all the resources (water, sand) and persons in deposit.
- `Person`: If visible, person name, carried or not status, dropped-off or not status.
- `Agent inventory`: List of all the resources (water, sand) and person (being carried) in the agent's inventory.

**Local Observations** $\mathcal{O}_L$ consists of all the objects that are visible in grid cells adjacent to the agent:

- For all the directions `<direction>` in (Up, Down, Left, Right, and Center), output of the following.

- `<direction>`: At direction `<direction>`, either 'Empty' if there is no object, 'Flammable' along with corresponding intensity & fire name if object is a part of a fire, or 'Obstacle' for any other object.

**Names** $\mathcal{O}_N$ consists of a list of the name of all visible and interactable objects.

## D.5 Action Space

The action space $\mathcal{A}$ consists of navigation actions $\mathcal{A}_{NAV}$, interaction actions $\mathcal{A}_{INT}$, exploration action $\mathcal{A}_{EXP}$.

**Navigation actions** $\mathcal{A}_{NAV}$ consists of the following actions:

- `Move(<direction>)`: Moves the agent to the neighboring grid cell in the specified direction where `<direction>` can be one of (`Up, Down, Left, Right, and Center`)
- `NavigateTo(<targetID>)`: Moves the agent next to the location of the object `<targetID>` if `<targetID>` is visible.

**Interaction actions** $\mathcal{A}_{INT}$ consists of the following actions:

- `Carry(<person>)`: Makes agent carry a person `<person>` if `<person>` is visible and interactable. The person is successfully 'group' carried, if at least the required number of agents successfully does the 'Carry(<person>)' action. If carry action is successful, all other resources in agent's inventory are dropped.
- `DropOff(<person>, <deposit>)`: Drops off person `<person>` at location `<deposit>`. This action is only successful if the person has been 'group' carried, all the agents carrying the person have the deposit be visible and interactable, and after all the agents do this action.
- `StoreSupply(<deposit>)`: Stores all the resources from the agent's current inventory in the deposit `<deposit>`.
- `UseSupply(<fire>, <supply-type>)`: Uses all the supplies of type `<supply-type>` on the fire `<fire>` at the location the agent is at.
- `GetSupply(<deposit>, <supply-type>)`: Fills the remaining of the agent's inventory space with the available supply of type `<supply-type>` in `<deposit>`.
- `GetSupply(<reservoir>)`: Collects 1 unit of supply from reservoir `<reservoir>` and stores it in the agent's inventory.

**Exploration action** $\mathcal{A}_{EXP}$ consists of the following actions:

- `Explore()`: Takes the agent in a direction of exploration as described by the heuristic exploration function described in algorithm 2.

---

**Algorithm 2** SAR Exploration Heuristic

---

**Input**: Agent $A$, Environment $env$, Previous Direction $D$, Current Position $P$, Max steps $M$
**Initialize**: New direction $ND \leftarrow \emptyset$

1: $ND \leftarrow$ randomly choose an angle from $[0, 2\pi)$ at $\pi/4$ intervals
2: **while** $ND = D$ **or** $ND \equiv D + \pi \mod 2\pi$
3:      **or** a barrier exists at most $\frac{3}{4}M$ steps in direction $ND$ from $P$ **do**
4:    $ND \leftarrow$ randomly choose an angle from $[0, 2\pi)$ at $\pi/4$ intervals
5: **end while**
6: Move agent $A$, $M$ steps in direction $ND$ in environment $env$

---

# E Pseudocode for LLaMAR

---

**Algorithm 3** LLaMAR

---

**Input**: $N$ agents, Task instruction $\mathcal{I}$, Environment $env$
**Initialize**: Memory $\mathcal{M} \leftarrow \emptyset$; Open Subtasks $\mathcal{G}_O \leftarrow \emptyset$;
Completed Subtasks $\mathcal{G}_C \leftarrow \emptyset$; Actions $a \leftarrow \emptyset$;
Corrective Actions $a_c \leftarrow \emptyset$
Actions Executed $d \leftarrow \emptyset$

1:  $o = (o_1, \cdots, o_N) = env.reset()$
2:  **while** $t < T$ **do**
3:      $\mathcal{G}_O \leftarrow \text{Planner}(\mathcal{I}, o, \mathcal{G}_O, \mathcal{G}_C, \mathcal{M})$
4:      $a, \mathcal{M} \leftarrow \text{Actor}(\mathcal{I}, o, a_c, \mathcal{G}_O, \mathcal{G}_C, \mathcal{M})$
5:      $o = (o_1, \cdots, o_N), d = (d_1, \cdots, d_N) = env.step(a)$
6:      $a_c \leftarrow \text{Corrector}(\mathcal{I}, o, a, d, \mathcal{G}_O, \mathcal{G}_C, \mathcal{M})$
7:      $\mathcal{G}_C \leftarrow \text{Verifier}(\mathcal{I}, o, a, d, \mathcal{G}_O, \mathcal{G}_C, \mathcal{M})$
8:      **if** $\mathcal{G}_O = \emptyset$ **then**
9:          **break**
10:      **end if**
11:      $t \leftarrow t + 1$
12: **end while**

---

# F Full Results

| Algorithm | LM | Success Rate | Transport Rate | Coverage | Balance | Steps |
|---|---|---|---|---|---|---|
| Act | GPT-4V | 0.33 (0.19, 0.49) | 0.67 (0.59, 0.76) | 0.91 (0.86,0.95) | 0.59 (0.52, 0.66) | 24.92 (22.12,27.73) |
| ReAct | GPT-4V | 0.34 (0.20, 0.49) | 0.72 (0.63,0.80) | 0.92 (0.86, 0.97) | 0.67 (0.61, 0.73) | 24.08 (21.27, 26.89) |
| CoT | GPT-4V | 0.14 (0.06, 0.28) | 0.59 (0.51, 0.67) | 0.87 (0.81, 0.92) | 0.62 (0.56,0.69) | 28.40 (26.91, 29.97) |
| SmartLLM | GPT-4V | 0.11 (0.05, 0.23) | 0.23 (0.13, 0.31) | 0.91 (0.80, 0.96) | 0.45 (0.37, 0.52) | 29.87 (26.20, 30.00) |
| CoELA | GPT-4V | 0.25 (0.10, 0.36) | 0.46 (0.35, 0.56) | 0.76 (0.67, 0.85) | 0.73 (0.67, 0.80) | 28.93 (27.77,30.00) |
| LLaMAR | GPT-4 | 0.51 (0.36, 0.66) | 0.85 (0.80, 0.91) | 0.95 (0.91, 0.98) | 0.83 (0.78, 0.86) | 25.80 (23.72, 27.88) |
| LLaMAR | LLaVA | 0.54 (0.41, 0.65) | 0.84 (0.71, 0.90) | 0.91 (0.87, 0.98) | 0.75 (0.64, 0.83) | 26.21 (21.56, 28.97) |
| LLaMAR | IDEFICS-2 | 0.57 (0.43, 0.67) | 0.86 (0.74, 0.91) | 0.94 (0.89, 0.98) | 0.78 (0.65, 0.84) | 25.27 (20.14, 28.37) |
| LLaMAR | CogVLM | 0.61 (0.47, 0.68) | 0.89 (0.73, 0.95) | 0.95 (0.89, 0.99) | 0.80 (0.73, 0.86) | 23.21 (20.57, 26.82) |
| LLaMAR (w/o exploration) | GPT-4V | 0.62 (0.46, 0.76) | 0.87 (0.80, 0.93) | 0.95 (0.91, 0.98) | 0.82 (0.77, 0.87) | 23.44 (20.88, 26.00) |
| LLaMAR (w/ exploration) | GPT-4V | **0.66 (0.50, 0.78)** | **0.91 (0.81,0.96)** | **0.97 (0.93,0.99)** | **0.82 (0.75,0.87)** | **21.87 (18.76,24.23)** |

Table 5: Comparison of evaluation metrics against baselines averaged across all tasks for the 2-agent MAP-THOR scenarios.

| Modules Used | Success Rate | Transport Rate | Coverage | Balance | Steps |
|---|---|---|---|---|---|
| Actor | 0.33 (0.19, 0.49) | 0.67 (0.59, 0.76) | 0.91 (0.86,0.95) | 0.59 (0.52, 0.66) | 24.92 (22.12,27.73) |
| Planner + Actor + Verifier | 0.45 (0.29, 0.57) | 0.78 (0.67, 0.84) | 0.92 (0.84, 0.95) | 0.69 (0.61, 0.75) | 24.87 (20.48, 27.95) |
| Planner + Actor + Corrector[‡] | **0.67** (0.51, 0.80) | **0.91** (0.83, 0.96) | **0.97** (0.94, 0.99) | **0.84** (0.79, 0.89) | 22.81 (19.95, 25.76) |
| LLaMAR | 0.66 (0.50, 0.76) | **0.91** (0.81, 0.96) | **0.97** (0.93,0.99) | 0.82 (0.75, 0.87) | **21.87** (18.76, 26.43) |

Table 6: Ablating different modules LLaMAR with GPT-4V as the underlying VLM, 2-agents scenarios.

| # of agents | MAP-THOR | | | | | SAR | | | | |
|---|---|---|---|---|---|---|---|---|---|---|
| | Success Rate | Transport Rate | Coverage | Balance | Steps | Success Rate | Transport Rate | Coverage | Balance | Steps |
| 1 | 0.37 (0.21, 0.51) | 0.67 (0.58, 0.74) | 0.87 (0.81, 0.90) | 1.00 (1.00,1.00) | 28.44 (25.23, 30.00) | 0.28 | 0.75 | 0.86 | **1.00** | 28 |
| 2 | 0.62 (0.46, 0.76) | 0.87 (0.80, 0.93) | 0.95 (0.91, 0.98) | 0.82 (0.77,0.87) | 23.44 (20.88, 26.00) | 0.44 (0.24, 0.65) | 0.86 (0.79, 0.94) | 0.94 (0.88, 0.99) | 0.91 (0.88, 0.95) | 27.76 (24.15, 30) |
| 3 | 0.70 (0.55, 0.82) | 0.91 (0.85, 0.95) | 0.98 (0.95, 0.99) | 0.66 (0.61, 0.71) | 21.30 (18.60, 23.99) | 0.68 (0.46, 0.85) | 0.92 (0.86, 0.98) | 0.96 (0.91,1.0) | 0.80 (0.73, 0.86) | 21.88 (17.83, 25.92) |
| 4 | 0.68 (0.52, 0.79) | 0.90 (0.84, 0.94) | 0.99 (0.95, 0.99) | 0.62 (0.57, 0.68) | 22.83 (19.63, 25.69) | 0.72 (0.50, 0.85) | 0.94 (0.88,0.98) | 0.98 (0.93, 1.00) | 0.78 (0.74, 0.83) | 22.00 (17.96, 26.03) |
| 5 | 0.62 (0.46, 0.75) | 0.90 (0.85, 0.94) | 0.99 (0.97,1.00) | 0.54 (0.48, 0.59) | 22.91 (20.26, 25.57) | **0.74** (0.52, 0.86) | **0.96** (0.94, 0.99) | **1.00** (1.0,1.0) | 0.73 (0.67, 0.79) | 24.52 (20.24,28.79) |

Table 7: LLaMAR with more agents

# G    Failure Cases

One of the main motivations of our paper was to achieve better success rates in planning for a variety of tasks compared to previous approaches. While our method outperforms other baselines, we acknowledge that the success rate is still under the expectation for real-world deployment. We believe that our approach LLaMAR and the MAP-THOR benchmark would serve as a starting point for future research to improve the success rates in multi-agent embodied robotics tasks. Here we describe a few major types of failures in our experiments:

- **Mis-generalization**: The agent can sometime fail to properly infer the level of abstraction to perform a task at, even if it should be clear from the action space. For example, for the washing tasks, the LM assumes that it must put the objects in the sink and add some, rather than just using the "CleanObject" action. We observe this error in the direction of performing actions more low-level than necessary, and not the other way around (more high-level than is feasible).
- **Mutual Interference (limited spatial reasoning)**: The agents sometimes block each other and thus fail to carry out actions such as placing an object on a receptacle, opening an object, etc. In particular, we see this behavior in the "put objects on the sofa" and the "put objects in a box" tasks, where the LM does not prevent the agents from blocking each other.
- **Improper Object Visibility (un-observability bias)**: The LM often fails to prioritize exploration for tasks that require objects not yet seen, simply due to it no being able to see it. This bias causes it to improperly assume that further exploration is not necessary, so it fails to find relevant objects.
- **SentenceBERT mismatch**: The free-form natural language output from the Actor is incorrectly mapped to the executable action. Based on our experimental results, 96.7% of the time the free-form natural language was correctly mapped to the feasible actions. The 2 main reasons for failures in the few cases it failed were:
  - **Incorrect mapping of objects**: When the agents have not explored the environment enough and the Actor module suggests the agents interact with objects that are not yet available in the agents' combined memory. Example: The free-form output "navigate to the table" was mapped to the action "NavigateTo(ArmChair)".
  - **Wrong object numeration**: When there are multiple objects of the same type in the memory, the sentenceBERT maps "Cabinet_3" to "Cabinet_1" which sometimes leads to incorrect actions being executed for the completion of the plan.
- **Limited horizon**: While each task is solvable by a single-agent human-controlled agent within our chosen episode length of $L = 30$, this is under the assumption of no failures during control and maximum visibility. For the autonomous agents the visibility distance is restricted to 1.5 meters, so while a human could identify objects from afar (from the rendered images) the agents have to explore the environment under a stricter partial observability constraint.

We hypothesize that a higher episode length would lead to higher metrics, but the cutoff was chosen for computational budget considerations. As can be observed in figure 6, the coverage and transportation rate metrics in particular seem to plateau around our chosen episode length $L$, so $L = 30$ is appropriate.

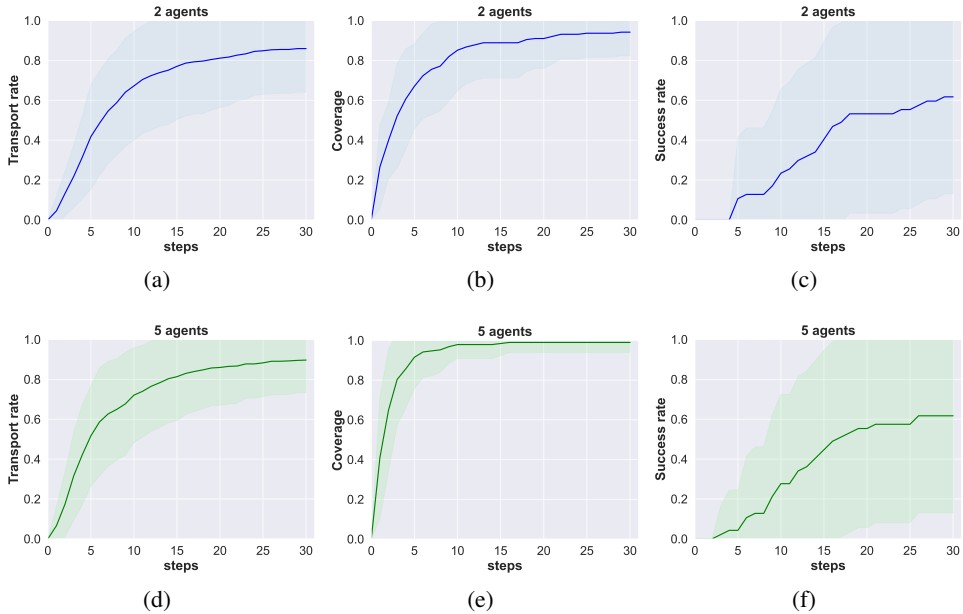

Figure 6: Plots of the average transport rate (a),(d); coverage (b),(e); and success rate (c),(f) metrics vs. maximum horizon steps for the GPT-4V, LLaMAR algorithm on MAP-THOR. Shown for both the 2 agent (a),(b),(c); and 5 agent cases (d),(e),(f), along with an error region of one standard deviation.

## G.1 Failure Cases

We have observed the following categories of failures for the LM on the SAR environment.

- **Incorrect Causal Ordering**: The LM sometimes fails to execute the steps in the proper causal ordering. For instance, we have observed the agent trying to drop off a lost person before picking them up, or properly getting supplies to stop the fire but not navigating to the fire location before using them.
- **Sub-Optimal Action Sequence**: Due to the time-constraints of the fire spread, it is not sufficient for the LM to do the sequence of tasks in order. They must also be done in a non-redundant, timely fashion. Otherwise, we have observed that the LM takes the right action sequence, but does not do it efficiently enough to stop the fire spread. For instance, if the agent prioritizes exploration to find the person, then the fire might spread until beyond a critical point where it cannot be extinguished within the horizon.
- **Catastrophic Failure Misunderstanding**: Even after the LM has completed the task, it can misunderstand a later failure as a cause for continuing the task. For instance, we have observed that if the LM has successfully dropped off a person, but then fails to pick them up again (which is expected since they're already dropped), it will think the task is incomplete.

## H   Baselines

While there are a lot of impressive LLM-based multi-agent planners as mentioned in Table 1, they vary in the assumptions about access to information about the environment. We were not able to find the official codebase for the Safe Multi-Agent Planning with Conformal Prediction [20] and TwoStep [21]. We describe the prompts used for our model as well as every baseline. Note that we show the prompt for the 2-agent case, but it is easily modified to generalize to the $n$-agent case. The italics and bolding added for emphasis.

### H.1   LLaMAR

We describe the prompts used for each of the modules used in LLaMAR:

## Prompt for Planner Module in LLaMAR

**You are an excellent planner** who is tasked with helping 2 embodied robots named Alice and Bob carry out a task. Both robots have a partially observable view of the environment. Hence they have to explore around in the environment to do the task.

You will get a description of the task robots are supposed to do. You will get an image of the environment from Alice's perspective and Bob's perspective as the observation input. To help you with detecting objects in the image, you will also get a list objects each agent is able to see in the environment. Here the objects are named as "<object_name>_<object_id>".
So, **along with the image inputs you will get the following information:**

### INPUT FORMAT ###
{**Task**: description of the task the robots are supposed to do,
**Alice's observation**: list of objects the Alice is observing,
**Bob's observation**: list of objects the Bob is observing,
**Robots' open subtasks**: list of subtasks the robots are supposed to carry out to finish the task. If no plan has been already created, this will be None.
**Robots' completed subtasks**: list of subtasks the robots have already completed. If no subtasks have been completed, this will be None.
**Robots' combined memory**: description of robots' combined memory}

Reason over the robots' task, image inputs, observations, open subtasks, completed subtasks and memory, and then output the following:
* **Reason**: The reason for why new subtasks need to be added.
* **Subtasks**: A list of open subtasks the robots are supposed to take to complete the task. Remember, as you get new information about the environment, you can modify this list. You can keep the same plan if you think it is still valid. Do not include the subtasks that have already been completed.
The "Plan" should be in a list format where the subtask are listed sequentially.
For example:
[*"locate the apple", "transport the apple to the fridge", "transport the book to the table"*]
[*"locate the cup", "go to cup", "clean cup"*]
When possible do not perform additional steps when one is sufficient (e.g. CleanObject is sufficient to clean an object, no other actions need to be done) Your output should be in the form of a python dictionary as shown below.

**Example output**:
{**"reason"**: *"Since the subtask list is empty, the robots need to transport the apple to the fridge and transport the book to the table.",*
**"plan"**: *["transport the apple to the fridge", "transport the book to the table"]*}

Ensure that the subtasks are not generic statements like "do the task". They should be specific to the task at hand.
Do not assign subtasks to any particular robot. Try not to modify the subtasks that already exist in the open subtasks list. Rather add new subtasks to the list.

* NOTE: DO NOT OUTPUT ANYTHING EXTRA OTHER THAN WHAT HAS BEEN SPECIFIED
Let's work this out in a step by step way to be sure we have the right answer.

## Prompt for Verifier Module in LLaMAR

**You are an excellent task verifier** who is tasked with helping 2 embodied robots named Alice and Bob carry out a task. Both robots have a partially observable view of the environment. Hence they have to explore around in the environment to do the task.

You will get a description of the task robots are supposed to do. You will get an image of the environment from Alice's perspective and Bob's perspective as the observation input. To help you with detecting objects in the image, you will also get a list objects each agent is able to see in the environment. Here the objects are named as "<object_name>_<object_id>".
So, **along with the image inputs you will get the following information**:

### INPUT FORMAT ###
{**Task**: description of the task the robots are supposed to do,
**Alice's observation**: list of objects the Alice is observing,
**Alice's state**: description of Alice's state,
**Alice's previous action**: the action Alice took in the previous step and if it was successful,
**Bob's observation**: list of objects the Bob is observing,
**Bob's state**: description of Bob's state, Bob's previous action: the action Bob took in the previous step,
**Robots' open subtasks**: list of open subtasks the robots in the previous step. If no plan has been already created, this will be None.
**Robots' completed subtasks**: list of subtasks the robots have already completed. If no subtasks have been completed, this will be None.
**Robots' combined memory**: description of robots' combined memory}

Reason over the robots' task, image inputs, observations, previous actions, open subtasks, completed subtasks and memory, and then **output the following**:
* **Reason**: The reason for why you think a particular subtask should be moved from the open subtasks list to the completed subtasks list.
* **Completed Subtasks**: The list of subtasks that have been completed by the robots. Note that you can add subtasks to this list only if they have been successfully completed and were in the open subtask list. If no subtasks have been completed at the current step, return an empty list.
The "Completed Subtasks" should be in a list format where the completed subtasks are listed.
For example: [*"locate the apple", "transport the apple to the fridge", "transport the book to the table"*]

Your output should be in the form of a python dictionary as shown below.

**Example output**:
{
**"reason"**: *"Alice placed the apple in the fridge in the previous step and was successful and Bob picked up the the book from the table. Hence Alice has completed the subtask of transporting the apple to the fridge, Bob has picked up the book, but Bob has still not completed the subtask of transporting the book to the table"*,
**"completed subtasks"**: [*"picked up book from the table", "transport the apple to the fridge"*]
}

* NOTE: DO NOT OUTPUT ANYTHING EXTRA OTHER THAN WHAT HAS BEEN SPECIFIED
When you output the completed subtasks, make sure to not forget to include the previous ones in addition to the new ones.
Let's work this out in a step by step way to be sure we have the right answer.

## Prompt for the Actor Module in LLaMAR

**You are an excellent planner and robot controller** who is tasked with helping 2 embodied robots named Alice, and Bob carry out a task. All 2 robots have a partially observable view of the environment. Hence they have to explore around in the environment to do the task.

They can perform the following actions:
[*"navigate to object <object_id>", "rotate in <rotation> direction", "pick up object <object_id>", "put object on <receptacle_id>", "open object <object_id>", "close object <object_id>", "slice object <object_id>", "toggle object <object_id> on", "toggle object <object_id> off", "clean object <object_id>", "look up by angle <angle>", "look down by angle <angle>", "move in <translation> direction", "stay idle", "Done"*]

Here *"Done"* is used when all the robots have completed the main task. Only use it when you think all the subtasks are complete.
*"stay idle"* is used when you want the robot to stay idle for a one-time step. This could be used to wait for the other robot to complete its subtask. Use it only when you think it is necessary.
Here *<rotation>* can be one of [*"Right", "Left"*].
Here *<angle>* is the angle in degrees and can only be one of [30, 60, 90, 120, 150, 180].
Here *<translation>* can be one of [*"Ahead", "Back", "Left", "Right"*].

So, **along with the image inputs you will get the following information**:

### INPUT FORMAT ###
{**Task**: description of the task the robots are supposed to do,
**Alice's observation**: list of objects the Alice is observing,
**Alice's state:** description of Alice's state,
**Alice's previous action**: description of what Alice did in the previous time step and whether it was successful,
**Alice's previous failures**: if Alice's few previous actions failed,
description of what failed,
**Bob's observation**: list of objects the Bob is observing,
**Bob's state**: description of Bob's state,
**Bob's previous action**: description of what Bob did in the previous time step and whether it was successful,
**Bob's previous failures**: if Bob's few previous actions failed, description of what failed,
**Robots' open subtasks**: list of subtasks supposed to carry out to finish the task. If no plan has been already created, this will be None.
**Robots' completed subtasks**: list of subtasks the robots have already completed. If no subtasks have been completed, this will be None.
**Robots' subtask**: description of the subtasks the robots were trying to complete in the previous step,
**Robots' combined memory**: description of robot's combined memory}

### OUTPUT FORMAT ###
First of all you are supposed to reason over the image inputs, the robots' observations, previous actions, previous failures, previous memory, subtasks and the available actions the robots can perform, and think step by step and then **output the following things**:

**\* Failure reason**: If any robot's previous action failed, use the previous history, your current knowledge of the room (i.e. what things are where), and your understanding of causality to think and rationalize about why the previous action failed. Output the reason for failure and how to fix this in the next timestep. If the previous action was successful, output "None".
Common failure reasons to lookout for include: one agent blocking another so must move out of the way, agent can't see an object or its destination and must explore (such as move, rotate, or look in a different direction) to find it, agent doing extraneous actions (such as drying objects when cleaning), etc. If the previous action was successful, output "None".

**\* Memory**: Whatever important information about the scene you think you should remember for the future as a memory. Remember that this memory will be used in future steps to carry out the task. So, you should not include information that is not relevant to the task. You can also include information that is already present in its memory if you think it might be useful in the future.

## (CONTINUED) Prompt for the Actor Module in LLaMAR

* **Reason**: The reasoning for what each robot is supposed to do next

* **Subtask**: The subtask each robot should currently try to solve, choose this from the list of open subtasks.

* **Action**: The actions the robots are supposed to take just in the next step such that they make progress towards completing the task. Make sure that these suggested actions make these robots more efficient in completing the task as compared to only one agent solving the task.
Your output should just be in the form of a python dictionary as shown below.

**Examples of output**:
**Example 1:**
{ **"failure reason"**: *"Bob failed to put the mug in the cabinet earlier because Alice was blocking it when she was putting the knife. To fix this, Alice should close the cabinet and move away , Charlie should move away to a different open area than Alice to avoid congestion, and Bob should wait until the next timestep until Alice can move aside."*,
**"memory"**: *"Alice finished putting the knife in the cabinet when Alice was at co-ordinates (1, .5) and was facing north. Bob wanted to put the mug in the cabinet when Bob was at co-ordinates (1, 0.25) and was facing north."*,
**"reason"**: *"Alice can close the cabinet door and then later back out in order help Bob with completing the task. Bob can be idle until the next timestep when Alice moves aside, by then Bob can navigate to the cabinet."*,
**"subtask"**: *"Alice is currently closing the cabinet door, Bob is currently waiting to get to navigate to the cabinet"*,
**"Alice's action"** : *"close the Cabinet_1"*,
**"Bob's action"** : *"stay idle"*
}

**Example 2**: {
**"failure reason"**: *"Bob failed to clean the cup earlier because Bob had not navigated to it, Bob assumed the cup to be in the sink which was erroneous. To fix this, Bob should navigate to the cup and in the next step clean cup."*,
**"memory"**: *"Alice finished navigating to the dish when Alice was at co-ordinates (-.5, .5) and was facing east. Bob was not able to clean the cup in the cabinet when Bob was at co-ordinates (1, .25) and was facing north."*,
**"reason"**: *"Alice can now clean the dish since Alice has navigated to it. Bob should navigate to the cup in order to be close enough to clean the cup."*,
**"subtask"**: *"Alice is currently trying to clean the dish, Bob is currently trying to navigate to the cup"*,
**"Alice's action"** : *"clean the dish object"*,
**"Bob's action"** : *"navigate to the cup"* }
Note that the output should just be a dictionary similar to the example outputs.

### Important Notes ###
* The robots can hold only one object at a time.
For example: If Alice is holding an apple, she cannot pick up another object until she puts the apple down.
* Even if the robot can see objects, it might not be able to interact with them if they are too far away. Hence you will need to make the robot navigate closer to the objects they want to interact with.
For example: An action like "pick up <object_id>" is feasible only if robot can see the object and is close enough to it. So you will have to navigate closer to it before you can pick it up.
* In some scenarios, the agents might not see the objects that they want to interact with. In such cases, you will have to make the robot explore the environment to find the object. In such scenarios you can use actions to rotate in place or look up / down or navigate to explore the environment.
* If you open an object, please ensure that you close it before you navigate to a different place.
* Opening object like drawers, cabinets, fridge can block the path of the robot. So open objects only when you think it is necessary.

* NOTE: DO NOT OUTPUT ANYTHING EXTRA OTHER THAN WHAT HAS BEEN SPECIFIED

## H.2 Act

We describe the prompt used for the Act baseline:

---

**Prompt for the Act Baseline**

**You are an excellent planner and robot controller** who is tasked with helping 2 embodied robots named Alice and Bob carry out a task. Both robots have a partially observable view of the environment. Hence they have to explore around in the environment to do the task.

They can perform the following actions:
[*"navigate to object <object_id>", "rotate in <rotation> direction", "pick up object <object_id>", "put object on <receptacle_id>", "open object <object_id>", "close object <object_id>", "slice object <object_id>", "toggle object <object_id> on", "toggle object <object_id> off", "clean object <object_id>", "look up by angle <angle>", "look down by angle <angle>", "move in <translation> direction", "stay idle", "Done"*]

Here *"Done"* is used when all the robots have completed the main task. Only use it when you think all the subtasks are complete.
*"stay idle"* is used when you want the robot to stay idle for a one-time step. This could be used to wait for the other robot to complete its subtask. Use it only when you think it is necessary.
Here *<rotation>* can be one of [*"Right", "Left"*].
Here *<angle>* is the angle in degrees and can only be one of [30, 60, 90, 120, 150, 180].
Here *<translation>* can be one of [*"Ahead", "Back", "Left", "Right"*].

You need to suggest the action that each robot should take at the current time step.

### **Important Notes** ###
* The robots can hold only one object at a time.
For example: If Alice is holding an apple, she cannot pick up another object until she puts the apple down.
* Even if the robot can see objects, it might not be able to interact with them if they are too far away. Hence you will need to make the robot navigate closer to the objects they want to interact with.
For example: An action like "pick up <object_id>" is feasible only if robot can see the object and is close enough to it. So you will have to navigate closer to it before you can pick it up.
* In some scenarios, the agents might not see the objects that they want to interact with. In such cases, you will have to make the robot explore the environment to find the object. In such scenarios you can use actions to rotate in place or look up / down or navigate to explore the environment.
* If you open an object, please ensure that you close it before you navigate to a different place.
* Opening object like drawers, cabinets, fridge can block the path of the robot. So open objects only when you think it is necessary.

### INPUT FORMAT ###
* You will get a **description of the task** robots are supposed to do.
* You will get an **image of the environment at the current time step** from Alice's perspective and Bob's perspective as the observation input. Here the objects are named as *"<object_name>_<object_id>"*.
* You will get a trace of the steps taken by the robots and the actions they took at each time step and whether it was successful or not.

### OUTPUT FORMAT ###
In your output, do not have any extra text or content outside of the python dictionary as below. Do NOT put any text, spaces, or enter keys (i.e. "/n") outside of it.

Your output should ONLY be in the form of a python dictionary, without any reasoning or extra text, as shown below:
{**"Alice"**: *"action to be taken by Alice"*,
**"Bob"**: *"action to be taken by Bob"*}

For example: If you think Alice should pick up an apple and Bob should navigate to the fridge, you will have to give the output as:
{**"Alice"**: *"pick up apple"*,
**"Bob"**: *"navigate to fridge"*}
* NOTE: DO NOT OUTPUT ANYTHING EXTRA OTHER THAN WHAT HAS BEEN SPECIFIED

---

## H.3 ReAct

We describe the prompt used for the ReAct baseline:

---

**Prompt for ReAct Baseline**

**You are an excellent planner** who is tasked with helping 2 embodied robots named Alice and Bob carry out a task. Both robots have a partially observable view of the environment. Hence they have to explore around in the environment to do the task.

They can perform the following actions: [*"navigate to object <object_id>", "rotate in <rotation> direction", "pick up object <object_id>", "put object on <receptacle_id>", "open object <object_id>", "close object <object_id>", "slice object <object_id>", "toggle object <object_id> on", "toggle object <object_id> off", "clean object <object_id>", "look up by angle <angle>", "look down by angle <angle>", "move in <translation> direction", "stay idle", "Done"*]
Here "Done" is used when all the robots have completed the main task. Only use it when you think all the subtasks are complete.
"stay idle" is used when you want the robot to stay idle for a one-time step. This could be used to wait for the other robot to complete its subtask. Use it only when you think it is necessary.
Here <rotation> can be one of [*"Right", "Left"*].
Here <angle> is the angle in degrees and can only be one of [30, 60, 90, 120, 150, 180].
Here <translation> can be one of [*"Ahead", "Back", "Left", "Right"*].

You need to suggest the action that each robot should take at the current time step.
### **Important Notes** ###
* The robots can hold only one object at a time.
For example: If Alice is holding an apple, she cannot pick up another object until she puts the apple down.
* Even if the robot can see objects, it might not be able to interact with them if they are too far away. Hence you will need to make the robot navigate closer to the objects they want to interact with.
For example: An action like "pick up <object_id>" is feasible only if robot can see the object and is close enough to it. So you will have to navigate closer to it before you can pick it up.
* In some scenarios, the agents might not see the objects that they want to interact with. In such cases, you will have to make the robot explore the environment to find the object. In such scenarios you can use actions to rotate in place or look up / down or navigate to explore the environment.
* If you open an object, please ensure that you close it before you navigate to a different place.
* Opening object like drawers, cabinets, fridge can block the path of the robot. So open objects only when you think it is necessary.
### **INPUT FORMAT** ###
* You will get a description of the task robots are supposed to do.
* You will get an image of the environment at the current time step from Alice's perspective and Bob's perspective as the observation input. Here the objects are named as "<object_name>_<object_id>".
* You will get a trace of the steps taken by the robots and the actions they took at each time step and whether it was successful or not.

### **OUTPUT FORMAT** ###
You are supposed to think and suggest the action each robot is supposed to take at the current time step. Before suggesting an action you need to think, which requires that you reason over the inputs and logically reflect on the task, observation and course of actions needed to complete the task.
Output Requirements: At each time step you must ONLY output a PYTHON DICTIONARY of the following two elements:
***First Element**: Key = **"Think"** | Value:(Type: String): A logical reflection of the best action to be taken given the inputs: task at hand, observations, and trace.
***Second Element**: Key = **"Action"** | Value:(Type: Python Dictionary):
The value should be in the form of a python dictionary as shown below.
{**"Alice"**: "action to be taken by Alice", **"Bob"**: "action to be taken by Bob"}

For example: If you think Alice should pick up an apple and Bob should navigate to the fridge, you will have to give the output as: {**"Alice"**: *"pick up apple"*, **"Bob"**: *"navigate to fridge"*}
Here is an **example output**:
{**"Think"**: *"To solve the task, I need to find and put the apple. The apple is likely to be on the countertop or table. Then find the fridge."*, **"Action"**: {**"Alice"**: *"pick up apple"*, **"Bob"**: *"navigate to fridge"*} }
* NOTE: DO NOT OUTPUT ANYTHING EXTRA OTHER THAN WHAT HAS BEEN SPECIFIED

---

## H.4 Chain of Thought

We describe the prompt used for the Chain-of-Thought baseline:

---

**Prompt for Chain of Thought Baseline**

**You are an excellent planner** who is tasked with helping 2 embodied robots named Alice and Bob carry out a task. Both robots have a partially observable view of the environment. Hence they have to explore around in the environment to do the task.

They can perform the following actions: [*"navigate to object <object_id>", "rotate in <rotation> direction", "pick up object <object_id>", "put object on <receptacle_id>", "open object <object_id>", "close object <object_id>", "slice object <object_id>", "toggle object <object_id> on", "toggle object <object_id> off", "clean object <object_id>", "look up by angle <angle>", "look down by angle <angle>", "move in <translation> direction", "stay idle", "Done"*] Here "Done" is used when all the robots have completed the main task. Only use it when you think all the subtasks are complete. "stay idle" is used when you want the robot to stay idle for a one-time step. This could be used to wait for the other robot to complete its subtask. Use it only when you think it is necessary. Here <rotation> can be one of [*"Right", "Left"*].
Here <angle> is the angle in degrees and can only be one of [30, 60, 90, 120, 150, 180].
Here <translation> can be one of [*"Ahead", "Back", "Left", "Right"*].

You need to suggest the action that each robot should take at the current time step.

### Important Notes ###
* The robots can hold only one object at a time. For example: If Alice is holding an apple, she cannot pick up another object until she puts the apple down.
* Even if the robot can see objects, it might not be able to interact with them if they are too far away. Hence you will need to make the robot navigate closer to the objects they want to interact with. For example: An action like "pick up <object_id>" is feasible only if robot can see the object and is close enough to it. So you will have to navigate closer to it before you can pick it up.
* In some scenarios, the agents might not see the objects that they want to interact with. In such cases, you will have to make the robot explore the environment to find the object. In such scenarios you can use actions to rotate in place or look up / down or navigate to explore the environment.
* If you open an object, please ensure that you close it before you navigate to a different place.
* Opening object like drawers, cabinets, fridge can block the path of the robot. So open objects only when you think it is necessary.

### INPUT FORMAT ###
* You will get a **description of the task** robots are supposed to do.
* You will get an **image of the environment at the current time step** from Alice's perspective and Bob's perspective as the observation input. Here the objects are named as "<object_name>_<object_id>".
* You will get a **trace of the steps taken by the robots** and the actions they took at each time step and whether it was successful or not.

### OUTPUT FORMAT ###
You are supposed to FIRST reason through the situation logically and step by step, then suggest the action each robot is supposed to take at the current time step.
In your output, do not have any extra text or content outside of the python dictionary as below.
Your output should ONLY be in the form of a python dictionary as shown below:
{**"reason"**: *"Reasoning for action plan...."*, **"Alice"**: *"action to be taken by Alice"*, **"Bob"**: *"action to be taken by Bob"*}
Put all of your reasoning inside of the "reason" key of the dictionary. Do NOT put any text, spaces, or enter keys (i.e. "/n") outside of it.

For example: If you think Alice should pick up an apple and Bob should navigate to the fridge, you will have to give the output as:
{**"reason"**: *"since the subtask list is empty, the robots need to transport the apple to the fridge"*, **"Alice"**: *"pick up apple"*, **"Bob"**: *"navigate to fridge"*}

Let's think step by step, but make sure to put all of your reasoning inside of the "reason" key of the dictionary!
* NOTE: DO NOT OUTPUT ANYTHING EXTRA OTHER THAN WHAT HAS BEEN SPECIFIED

---

### H.5 SmartLLM

We adapt the prompt from the official codebase of SmartLLM (`master` branch; commit #be42930050f7d4d8f2fad027aff14a699c3300aa) as given here: https://github.com/SMARTlab-Purdue/SMART-LLM/blob/master/scripts/run_llm.py with a slight modification. Instead of letting the agents access all the objects in the environment through the simulator metadata, we just give the list of objects visible from the agents' point-of-view.

### H.6 CoELA

We adapt the prompt from the official codebase of CoELA (`master` branch: commit #3d34de46dc77f9aaabe438cd2b92ea6c5c04973a) as given here: https://github.com/UMass-Foundation-Model/Co-LLM-Agents/tree/master/tdw_mat/LLM. We modify some aspects of the prompt as described: Instead of relying on the simulator/pre-defined conditional logic for generating the list of available action options, we give a list of all possible actions based on the observation. This includes the option to send the communication message, all navigation actions, and all combinations of valid actions with the interactable objects in the current observation.

## I  Open Source VLMs

We list the source of the weights we used for the open-source VLMs:

- **Idefics 2** [68, 69]: We use the 8B base model fine-tuned on a mixture of supervised and instruction datasets (text-only and multimodal datasets) from HuggingFace. The weights were downloaded from https://huggingface.co/HuggingFaceM4/idefics2-8b with the commit #2c031da2dc71f3ac989f9efa9b8ff476df3842c0. We chose Idefics because it is able to take multiple images as input similar to GPT-4V and reason on them.
- **LLaVA** [70]: We use the 7B model t trained by fine-tuning LLaMA/Vicuna on GPT-generated multimodal instruction-following data. The weights were downloaded from https://huggingface.co/llava-hf/llava-1.5-7b-hf with the commit # 05ae2434cbb430be33edcba0c5203e7023f785b7.
- **CogVLM** [72]: We use the 18B model. The weights were downloaded from https://huggingface.co/THUDM/cogagent-chat-hf with the commit # d519da3b191401234f4bd86ce1c287c61bc276a3.

## J  SentenceBERT fine-tuning

We finetuned a pre-trained BERT model to function as a semantic mapper between free-form natural language output and the robot's admissible actions in the environment. The pre-trained weights were obtained from https://huggingface.co/sentence-transformers/all-MiniLM-L6-v2. The model was trained on a dataset consisting of 2800 free-form input, valid action output pairs. It ran on one (1) Apple M1 core for a wall clock time of 5 minutes. Table 8 shows the hyper-parameters used for the pre-training of the BERT model.

| | |
|---|---|
| Epochs | 10 |
| Max gradient norm | 1 |
| Learning rate | $2 \times 10^{-5}$ |
| Batch size | 64 |
| Encoding dimension | 384 |
| Optimizer | AdamW |
| Scheduler | Warm-up linear |
| Warm-up steps | 45 |
| Weight decay | 0.01 |
| Loss scale | 20 |
| Loss type | Multiple negatives ranking loss |
| Similarity function | Cosine similarity |

Table 8: Hyper-parameters for the model fine-tuning including the loss.

