# OpenReview forum: "Long-Horizon Planning for Multi-Agent Robots in Partially Observable Environments"
_NeurIPS.cc/2024/Conference — NeurIPS 2024 poster_

### Official Review · Reviewer_QApg · 2024-07-09

**Soundness:** 3
**Presentation:** 4
**Contribution:** 3
**Rating:** 6
**Confidence:** 4

**Summary:**

The paper introduces the Long-Horizon Planner for Multi-Agent Robotics (LLaMAR), a cognitive architecture leveraging Language Models (LMs) for planning tasks in partially observable environments. Unlike traditional methods, LLaMAR uses a plan-act-correct-verify framework, enabling real-time self-correction and task verification without relying on oracles or simulators. The system's effectiveness is demonstrated through the MAP-THOR benchmark, which includes various household tasks in the AI2-THOR environment. LLaMAR achieves a 30% higher success rate than other state-of-the-art LM-based multi-agent planners.

**Strengths:**

1. The division of tasks into specialized modules (Planner, Actor, Corrector, Verifier) allows for efficient task management and error correction.
2. The system can self-correct based on real-time feedback, reducing the reliance on perfect environmental knowledge or external oracles.
3. Experimental results show that LLaMAR achieves a significantly higher success rate compared to other LM-based planners, indicating its effectiveness.

**Weaknesses:**

1. As with most LLM-based TAMP work, how to scale to the real world remains a real concern.
2. The use of multiple LM-based modules for each decision step increases computational and time costs compared to simpler baselines.
3. Some other LLM-based planning framework also has a correction strategy [1]. This framework utilizes A* with symbol operators to verify the plan. Also, this strategy can ensure that the generated plan is the shortest, which is a limitation in this paper.
4. In the appendix, the number of actions used by these tasks is at most about 10. In complex real-world environments, LMs may need to select the required actions from hundreds of different actions according to different tasks. This complex framework may lead to problems such as difficulty in finding the optimal solution and high computational complexity.
5. Too many restrictions might limit the capabilities of LMs.

[1]. Li, Z., Yu, K., Cheng, S., & Xu, D. (2024). LEAGUE++: Empowering continual robot learning through guided skill acquisition with large language models. In ICLR 2024 Workshop on Large Language Model (LLM) Agents. Retrieved from https://openreview.net/forum?id=xXo4JL8FvV

**Questions:**

1. Do you have any strategies in mind for optimizing or reducing this computational overhead?

**Limitations:**

1. How to scale to the real world remains a real concern.
2. Computational complexity is a big concern in more complex environments.
3. Too many restrictions might limit the capabilities of LMs.

---

> ### Author Rebuttal · Authors · 2024-08-06
>
> Dear Reviewer,
>
> Thank you for your comprehensive review and insightful feedback on our paper. We appreciate your thoughtful comments and suggestions, and we have provided our responses and clarifications below.
>
> ### Computational complexity
>
> Yes, we agree that having multiple LM-based modules would increase the computational complexity of the planning. Our initial goal was to create a benchmark and a method for getting the multi-agent robot teams to complete the tasks with a higher success rate as compared to previous approaches and hence it necessitated the need for more modules. While our initial experiments (refer to Table 2) compared the performance of different open-source along with GPT-4v, we found that GPT-4v was the best-performing one. Our future work would include using specialized fine-tuned LM modules. This would allow us to use smaller LMs and decrease the computational resources required to utilize all 4 modules.
>
> ### Number of actions
>
> While our action space contains the following executable actions: Move(<direction>), Rotate(<direction>), LookUp(<angle>), LookDown(<angle>), NavigateTo(<object_id>), Pickup(<object_id>), Put(<receptacle_id>), etc. There are a lot of different combinations available where the <object_ids>, <directions>, and <angles> can change. Hence the action space inherently becomes combinatorially large. For this reason, we do not constrain the LM modules to choose actions in a fixed format from this set; rather, we obtain a free-form language output. This allows for more expressivity in the outputs. The sentenceBERT module maps these free-form natural language outputs to the executable actions in the environment.
>
> ### Scaling to real-world settings
>
> We agree that scaling LLM-based TAMP methods is a challenge. We perform experiments with LLaMAR in a multi-agent drone search-and-rescue environment (SAR). Refer to the global rebuttal. The idea of having these experiments was to test out the capabilities of LLaMAR in other multi-agent environments. While the metrics reported with LLaMAR are in a simulator, we are hoping to run more real-world experiments with these drones to test how they fare when executed in a more realistic setting.
>
> ### League++ Reference
>
> Thanks a lot for bringing this very recent paper (April 2024) to our attention. While League++ does tackle the TAMP problem in conjunction with reinforcement learning a few key differences to our work are:
>
> 1. They focus on single-agent scenarios whereas we focus on multi-agent planning
> 2. They use LLMs for reward generation and skill acquisition.
> 3. The agent is given a list of all objects as a part of the observation which allows for creating plans and performing an A* search to verify the plan. In our case, we have a partially observable view of the environment which does not allow for performing search over the possible tasks/actions.
>
> ### Challenge posed by over-restricted environments
>
> Having over-constrained systems in various robotics settings is a well-known challenge (eg., freezing-robot problem) that could lead to issues in scaling. We agree with the comment on how our proposed method is also limited in similar settings. We will add this in our limitations section in the paper.

---

> > ### Comment · Reviewer_QApg · 2024-08-12
> > **Thanks for your response!**
> >
> > Thanks for the rebuttal! I appreciate the explanations for the action space and add some limitations. Although this framework has not been proven to be used in real world, it has potential to solve this challenging problem.

---

> > > ### Author Response · Authors · 2024-08-12
> > >
> > > Dear reviewer QApg,
> > >
> > > Thank you for acknowledging the rebuttal. We are hoping to perform some real-world experiments, especially in the search-and-rescue setting. We appreciate your time and efforts in reviewing our paper. Your feedback has been valuable and we will incorporate your suggestions in the final paper.

---

### Official Review · Reviewer_snqg · 2024-07-11

**Soundness:** 3
**Presentation:** 3
**Contribution:** 3
**Rating:** 6
**Confidence:** 2

**Summary:**

This paper presents a multi-agent planning and execution engine called LLaMaR that is based on several multi-modal foundation models such as GPT4V, i.e., large models  (LMs) that can reason about text and visual inputs. LLaMaR has 4 LMs: planner, actor, corrector, and verifier. The planner decomposes the task into subgoals. The actor chooses which subgoal to perform and which low-level actions to perform to achieve it. The corrector modifies the actions chosen by the actor in case the chosen action is known to have failed. The verifier verifies which subgoals have been achieved. No additional knowledge is given to LLaMaR. LLaMaR controls all agents in a centralized manner.
The paper also introduce briefly a benchmark for evaluating LLaMaR which is based on the AI2Thor simulator, and shows that LLaMaR outperforms similar LM-based planners. The main novelty in LLaMaR is the corrector module, and the fact that it is designed for partially observable situations. The latter is addressed by allowing the agents to invoke an exploration strategy if the object they seek is unavailable.

**Strengths:**

1.	The problem is extremely challenging, and the proposed approach works well on the benchmarks presented in this work.
2.	The topic is highly relevant.
3.	The notion of a corrector as a means of learning from mistakes is interesting.
4.	Having exploration actions to discover unobserved objects is also a nice contribution.
5.	The authors present a new benchmark for evaluating algorithms on this problem. This will help future research.

**Weaknesses:**

1.	While the title promises some multi-agent aspect, I didn’t see anything in LLaMaR that is specific for multi-agent planning. What about interactions between actions? communication issues? Or decomposition of the action-space for efficiency purposes?
2.	The analysis of Table 5 is, as I see it, incorrect. See questions below.
3.	LLaMaR is described in a hand-wavy and shallow manner.
4.	The exploration strategy seems ad-hoc and domain specific.

**Questions:**

1.	The separation of the Actor and the Corrector seems artificial. What does the Actor know that the Corrector does not? Since the Corrector needs to be smart enough to know the subgoal, know past failures, and suggest a course of action, it seems it has all the input that the Actor has and more.
2.	What elements of LLaMaR are really multi-agent? The agents are sharing information in a seamless way, and there is nothing LLaMaR does to avoid the combinatorial challenge of planning for multiple agents (huge number of actions, interactions between the agents).
3.	You say that you didn’t use the Smart-LLM benchmark since some of the tasks did not support multiple agents. Wouldn’t it be good to at least use the tasks there that do support multiple agents?
4.	With respect to Table 5 and the impact of adding more agents, you wrote that the success rate “remains high” and the number of steps “generally decreases with more agents”. Yet when moving from 3 to 4 agents and from 4 to 5 agents, the success rate drops and the number of steps increases. Is the conclusions drawn from Table 5 indeed incorrect?

**Limitations:**

Not relevant

---

> ### Author Rebuttal · Authors · 2024-08-06
>
> Dear Reviewer,
>
> Thank you for your comprehensive review and insightful feedback on our paper. We appreciate your thoughtful comments and suggestions, and we have provided our responses and clarifications below.
>
> ### LLaMAR explanation
>
> The specific order of the modules is due to natural causal relationships in which environment feedback is received, which restricts our freedom in permuting the order significantly.
>
> We use the planner as the first module because it allows LLaMAR to come up with an initial list of open subtasks that could be completed based on the current observation to satisfy the task. This list serves as a rough high-level plan. The actor then uses this information to suggest the necessary actions.
>
> The Corrector is used after the Actor module so as to check for any failures in the execution of the actions suggested by the Actor. Note that the failure module is inert and only suggests corrective actions. Only the Actor module has the final say in the actions that are supposed to be executed. This distinction allows for clear reasoning on failures when they occur and lets the actor module focus on choosing actions.
>
> The verifier is used after the action is executed to update the list of closed subtasks so that LLaMAR can be current with the progress toward the completion of the task. This allows the planner to update the list of open subtasks in the next step. In essence, the planner and the verifier ensure that the progress of the agents is tracked and the actor and the corrector ensure that the actions are executed successfully to advance towards completion of the task.
>
> Our revised manuscript will include this justification with illustrative examples so as to explain why we converged on this solution.
>
> ### Multi-agent features in LLaMAR
>
> - **Coordination Through Communication:** Agents share their state information with the centralized LLaMAR modules to predict actions, enabling them to coordinate and avoid conflicts. This information sharing allows for the agents to cooperate and achieve the collective goal. A limitation of having a centralized module is that as the number of agents increases the observation input size (both textual and visual features) increases leading to a large input query (prompt).
> - **Hierarchical Task Decomposition:** LLaMAR decomposes complex tasks into subgoals that are distributed among agents. This hierarchical approach simplifies the planning process and reduces computational complexity by focusing on manageable sub-tasks.
> - **Dynamic Role Assignment:** Agents are dynamically assigned roles based on the current task requirements and their capabilities. This flexibility allows LLaMAR to adapt to changing environments and task demands.
> - **Decomposition of Action-Space:** To handle the complexity of multi-agent planning, LLaMAR decomposes the action space by creating specific subgoals/subtasks available for any agent to assign itself (done by the actor module) based on the observation & current context. This decomposition reduces the overall search space and improves planning efficiency.
>
> ### The corrector and actor module separation
>
> Correct, the actor and corrector have similar information as inputs, but they are queried with a different task prompt. That is, we task the Actor module to suggest the actions and the Corrector module to explicitly reason why certain actions failed. This explicit designation of tasks in LM agents is more effective [1] as compared to letting a single LM module predict both the actions and reason for failure. Finally, only the ‘actor’ LM has the final say in the action to be performed, and the ‘corrector’ module is inert, only serving as a reasoning tool for the ‘actor’ LM.
>
> [1] Guo, T., Chen, X., Wang, Y., Chang, R., Pei, S., Chawla, N. V., ... & Zhang, X. (2024). Large language model based multi-agents: A survey of progress and challenges. *arXiv preprint arXiv:2402.01680*.
>
> ### Table 5 Analysis
>
> In Table 5, there is a decrease in the metrics when the number of agents increases from 3 to 5. The reason for this is that as the number of agents increases, the rooms in MAP-THOR get crowded. Hence the agents block each other while navigating in the environment leading to a decrease in success rates due to successive failures in navigating to desired locations. We will update the analysis for Table 5 in our revised manuscript to explain the decrease in the metrics as we increase the number of agents. We apologize for the confusion! We perform experiments with LLaMAR in a multi-agent drone search-and-rescue environment (SAR) to showcase the effectiveness of LLaMAR in a different environment. Please refer to the global rebuttal for results and analysis.
>
> ### Exploration Strategy
>
> We agree that the exploration strategy used is domain-specific as we wanted the agents to search in more semantically related regions for objects of interest. We hope to use more sophisticated & generic approaches for exploration in our future work such that can be generalized well to other scenarios.
>
> ### SmartLLM Benchmark
>
> MAP-THOR is designed to include tasks that could be done by multiple agents. SmartLLM includes many tasks that could be completed with just a single agent. Example: “Slice the tomato”, “throw the spatula in the trash”, etc. [Link](https://github.com/SMARTlab-Purdue/SMART-LLM/blob/master/data/final_test/FloorPlan6.json). Although many of the tasks in these benchmarks require single agents, there are a few tasks that could be solved with multiple agents. We include most of such tasks in our benchmark. We will ensure that we include any other tasks from the SmartLLM benchmark that are not present in MAP-THOR.

---

> > ### Comment · Reviewer_snqg · 2024-08-12
> > **Thanks! some follow up concerns**
> >
> > Thanks for the rebuttal!
> > I appreciate the justification for the components' execution order and the justification for the Corrector.
> > Re. the multi-agent aspect I admit I'm not fully convinced. It seems this paper could have been written for a single-agent problem without any changes to the theory. Having all agents share observations means this is very close to a single-agent situation. In general, I feel the multi-agent aspect of this paper is not fully developed.

---

> > > ### Author Response · Authors · 2024-08-13
> > >
> > > Thanks a lot for acknowledging the rebuttal! We try to elaborate on the multi-agent aspect of our paper below:
> > >
> > > The ease of scaling our pipeline from the single to the multi-agent setting shows that our approach is generalizable enough to encompass these two settings without having to distinguish the two. There were still many non-trivial roadblocks to naively scaling to multiple agents that we had to address.
> > >
> > > The core of our architecture relies on the planner & verifier after each iteration to output and update, respectively, a globally accessible set of open and closed high-level subtasks to accomplish a task. Reasoning from memory, the environment state, and the aforementioned list, the actor & corrector modules output the appropriate actions for each agent. Thus, the actor & corrector modules have flexibility in the how of action execution, while the planner & verifier have over the what.
> > >
> > > Here we quickly discuss how our architecture is able to handle challenges that naturally arise from a multi-agent setting.
> > > - Task allocation & environment non-stationarity
> > >     - Our architecture is able to distinguish between the deconstruction of a task into subgoals required for long-term planning, and the details of action execution such as low-level causal dependencies (e.g. grab object before storing), assigning specific responsibilities to agents, etc.
> > >     This allows for detours, self-correction, and changes in responsibilities between agents on-the-fly, while not compromising long-term planning and focused task completion.
> > >
> > > These considerations are of pressing importance in the multi-agent setting especially where there are collisions/interference and heavy non-stationarity in the environment state.
> > >
> > > - Computational Complexity
> > >     - To resolve the problem of increasing computational complexity in long-term planning, we opt for on-the-fly action responses (done by the actor module). Namely, we avoid generating the entire sequence, or even a subset, of each agent's actions through time. This avoids the challenges of ensuring non-interference & proper collaboration far into the future. These apriori planning approaches that might work in the single-agent setting would lead to untameable complexity as more agents are added.
> > >
> > > Our pipeline, we found, was a flexible solution that gave a structure for the LM to reason about multiple agents without explicitly doing so.
> > >
> > > We very much agree that the addition of asynchrony or decentralization (not sharing observations) in planning poses a greater challenge and is a fruitful direction for future research.

---

### Official Review · Reviewer_XzUd · 2024-07-12

**Soundness:** 3
**Presentation:** 2
**Contribution:** 2
**Rating:** 5
**Confidence:** 3

**Summary:**

This paper focuses on the long-horizon multi-agent planning tasks. They propose a new agent structure  composed of four language models (Planner, Actor, Corrector, Verifier), named Language-Model-based Long-Horizon Planner for Multi-Agent Robotics (LLaMAR). LLaMAR operates without prior knowledge or privileged information from the environment, making it suitable for practical robotic applications. Additionally, the authors develop a multi-agent planning benchmark upon the AI2-THOR simulator, encompassing 45 household tasks. Empirical results demonstrate that their method outperforms existing agent methods.

**Strengths:**

1. They propose a novel agent structure for building a centralized multi-agent system using (vision-)language models. The integration of the corrector and verifier modules enables the system to perform effectively in practical scenarios. An ablation study on different modules highlights the significant contributions and effectiveness of these two components.

2. The paper demonstrates clear and well-organized writing and includes extensive experiments that support their statements.

**Weaknesses:**

While the agent structure design is commendable, its contribution to the multi-agent setting appears limited. Firstly, the proposed benchmark focuses on evaluating the effectiveness and robustness of multi-agent planning, yet these tasks are not necessarily crucial for multi-agent cooperation. As shown in Table 5, increasing the number of agents does not lead to improvement and, in fact, results in a decline in almost all metrics when the number of agents increases from 3 to 5. Secondly, task allocation, which is critical for a centralized multi-agent system, is not thoroughly discussed in this paper.

Moreover, considering the overall performance is relatively poor, with at least one-third of the trials failing, more in-depth analyses of the failure cases would be expected.

**Questions:**

1. Since the agent structure can be easily adapted to a single-agent setting, a comparison between single-agent and multi-agent systems would be valuable to demonstrate the effectiveness of using a multi-agent system for these tasks.

2. I would expect more in-depth analyses of the failure cases, such as examining the behavior of different methods across various categorized tasks and identifying the causes of the significant gap between the baselines and the proposed method. Additionally, it would be helpful to understand the conditions under which the proposed method tends to fail.

3. Some visualized examples would be helpful for understanding how the agents cooperate to complete long-horizon tasks.

**Limitations:**

The authors have addressed the limitations of their work.

---

> ### Author Rebuttal · Authors · 2024-08-06
>
> Dear Reviewer,
>
> Thank you for your comprehensive review and insightful feedback on our paper. We appreciate your thoughtful comments and suggestions, and we have provided our responses and clarifications below.
> ### Single-agent comparison in MAP-THOR:
> We perform experiments with just a single agent in the environment for the task in MAP-THOR. We report all the metrics for the single agent setting below. We report the means and the 95% confidence interval in parentheses.
> | # agents | Success Rate | Transport Rate | Coverage | Balance | Steps |
> | --- | --- | --- | --- | --- | --- |
> | 1 | 0.37 (0.21,0.51) | 0.67 (0.58, 0.74) | 0.87 (0.81, 0.90) | 1.00 (1.00,1.00) | 28.44 (25.23,30.00) |
>
> With just a single agent in the environment, it is not able to complete the task in a majority of the episodes. This is because the number of steps required to complete the episode is higher than the horizon length of $L=30$. Hence the success rate and the transport rate are lower than n>1 agents. Also, the singular agent in the environment is not able to explore the environment enough within the planning horizon and hence the coverage is lower than $n>1$ agents.
>
> ### Multi-agentness in experiments
> We perform experiments in a multi-agent search-and-rescue environment where multiple drones are required to collaborate together. More information can be obtained in the global rebuttal. The key relevant points in this new environment are:
> - **Scalability**: With the addition of multiple parallel emergency disaster scenarios at once, the environment scales comfortably to more agents. Additionally, the complexity slowly increases as the model has to coordinate between more actors and competing emergencies.
> - **Explicit cooperation**:
>     - In order to move a group of humans, at least 2 agents are required. Thus, explicit multi-agent collaboration is required.
>     - Due to the time pressure, in order to successfully stop the fire, drones must all effectively collaborate in collecting and strategically using resources in high-intensity regions.
> - **Exploration task assignment**: There is an explicit tradeoff between allocating drones towards exploring to find lost personnel and fighting the current fires.
>
> ### Table 5 Analysis
> In Table 5, there is a decrease in the metrics when the number of agents increases from 3 to 5. The reason for this is that as the number of agents increases, the rooms in MAP-THOR get crowded. Hence the agents block each other while navigating in the environment leading to a decrease in success rates due to successive failures in navigating to desired locations. We will update the analysis for Table 5 in our revised manuscript to explain the decrease in the metrics as we increase the number of agents. We apologize for the confusion!
>
> We have performed experiments in a search-and-rescue environment that requires explicit cooperation. Please refer to the global rebuttal for the results in the search-and-rescue environment and more analysis.
>
> ### Task Allocation
> In our proposed architecture, the planner module plays the role of creating subtasks necessary for completing the tasks. The actor module then takes these subtasks and allocates them to different agents based on each agent's state. Centralized planning ensures that the actions suggested are not in conflict.
>
> ### Analysis of failures cases
>  Multi-agent planning in embodied robotics environments has been a challenging problem. Our initial goal was to achieve better success rates in planning for a variety of tasks compared to previous approaches. While our method outperforms other baselines, we agree that the success rate is still not as good as expected for real-world deployment. We believe that our approach LLaMAR and the MAP-THOR benchmark would serve as a starting point for future research to improve the success rates in multi-agent embodied robotics tasks.
>
> A few major types of failures in our experiments were:
> - **Mis-generalization**: For example, in the wash tasks the LM assumes that it must put the objects in the sink, and add soap; rather than using the “CleanObject” action.
> - **Mutual interference (limited spatial reasoning)**: For example, in the put objects on the sofa task, the LM allows the agents to block each other, and therefore fail to put the objects on the sofa. Similarly, for the put objects in a box task.
> - **Improper receptacle visibility**: For example, in the put objects on table task, the table is not initially visible yet the LM does not prioritize exploring to find it. Thus, it fails to navigate to it to put the necessary objects.
> - **SentenceBERT mismatch**: The free-form natural language output from the Actor is incorrectly mapped to the executable action. Based on our experimental results, 96.7% of the time the free-form natural language was correctly mapped to the feasible actions. The 2 main reasons for failures in the few cases it failed were:
>     1. Incorrect mapping of objects: when the agents had not yet explored the environment enough and the Actor module suggested the agents interact with objects that were not yet available in the agents’ combined memory. Example: The free-form output “navigate to the table” was mapped to the action “NavigateTo(ArmChair)”.
>     2. Wrong object numeration: When there are multiple objects of the same type in the memory, the sentenceBERT maps “Cabinet_3” to “Cabinet_1” which sometimes leads to incorrect actions being executed for the completion of the plan.
>
>     We will add a comprehensive list of different reasons for failures in each of the task categories in our revised appendix to help the readers better understand the gaps in performance between the baselines and our proposed approach. Along with this, we agree that having more illustrative examples of cooperation among agents in completing tasks would be helpful to showcase the effectiveness of our method. We will add these illustrations in our revised manuscript.

---

> > ### Comment · Reviewer_XzUd · 2024-08-11
> >
> > Thank you for your thorough response and the effort you've put into addressing my concerns. Based on the feedback and new results, I have a few follow-up questions:
> >
> > 1. Given that a 30-step episode may be insufficient for a single agent to complete the task, could you explain why this episode length was chosen? Additionally, considering the limited space in the current environments, which might hinder effective multi-agent cooperation, what would happen if the episode length were extended?
> > 2. The inclusion of additional search-and-rescue experiments seems to have complicated the organization of the MAP-THOR benchmark. Could you clarify the number of scenario types included in MAP-THOR and highlight the main differences between them? A table would be particularly useful to clearly present the structure of MAP-THOR and the rationale behind it.
> > 3. I appreciate your detailed feedback on the failure cases. To better guide future improvements, more quantitative analysis would be helpful. For instance, similar to Fig 4 in [1], such analysis could clarify the impact of the proposed modular system on overall performance and identify the most critical modules.
> >
> > [1] Liu, Peiqi, et al. "Ok-robot: What really matters in integrating open-knowledge models for robotics." in RSS 2024.

---

> ### Author Response · Authors · 2024-08-12
> **Part 1**
>
> Thanks a lot for your reply and your follow-up questions!
>
> 1. We chose $L=30$ as our episode length for computational and budget reasons (to limit the GPT-4 API costs) to fit all the experiments we wanted to do and hence we did not extend the horizon. However, we ensured that all of the tasks were solvable by a single-agent human-controlled agent. The major difference between a human-controlled agent and an autonomous agent is that the visibility distance is set to 1.5m for the autonomous agents whereas it is not restricted for a human-controlled agent. A human can identify objects from quite far away (from the rendered images) and hence are able to efficiently navigate in the environment to interact with the objects whereas autonomous agents have to explore the environment. Thanks a lot for bringing up this point. We apologize for not including these details in the appendix.
> 2. We agree that a table clarifying the different scenario types in MAP-THOR would help the readers understand the rationale behind having them. Below is a description of the type of tasks in MAP-THOR along with the differences.
>     1. Explicit item type, quantity, and target location (Type 1): Since the name and the quantity of the items to be interacted with and their exact target locations are explicitly defined, these tasks allow the agents to come up with an initial plan even without needing the agents to explore the environment.
>
>         Example: the task “put bread, lettuce, and fridge in the fridge” can be solved by creating an initial plan:
>
>         - transport bread to fridge
>         - transport lettuce to fridge
>         - transport bread to fridge
>
>         These types of tasks are used in the experiments in the CoELA and SmartLLM paper.
>
>     2. Explicit item type and target location but implicit item quantity (Type 2): Similar to Type 1, since the name of the items to be interacted with and their target locations are known, the agents can come up with an initial plan. But here, the number of such objects is not known apriori. Hence the agents will have to explore the environment to identify all such instances of these objects.
>
>         Example: the task “put all vases on the table” can be partially solved by creating an initial plan:
>
>         - transport vase to the table
>         - explore the environment to search for more vases
>     3. Explicit target location but implicit item types and quantity (Type 3): In these type of tasks, the agents cannot create an initial plan to solve the task as easily as Type 1 and Type 2. Here the names of the objects that it is supposed to interact with are not known apriori. The agents need to explore the environment such that the planner module can add subtasks that need to be satisfied.
>
>         Example:  for the task “put all groceries in the fridge”, the agents have to first identify the relevant objects that fit the category of “groceries” and then add subtasks to interact with those objects appropriately. This showcases the effectiveness of the planner module, verifier module, and exploration module to update the open and closed subtasks list by efficiently exploring the environment.
>
>     4. Implicit target location, item type, and quantity (Type 4): This is similar to Type 3 tasks where the objects it needs to interact with are not known apriori. The key difference is that the target locations are also not known in this case whereas it is explicitly defined in Type 4 tasks. This checks whether the underlying LM is able to identify the appropriate target locations for the objects that it needs to interact with.
>
>         Example: “Clear the table by placing the items in their appropriate positions”. In this task, the LM is supposed to first identify the objects on the table and their appropriate target locations.
>
>         - transport bread to the fridge,
>         - transport apple to the fridge
>         - transport tomato to the fridge
>         - transport knife to the cabinet/countertop/drawer
>         - transport bowl to the cabinet
>         - transport book to the bookshelf
>
>         Note that the ambiguity of the tasks increases in each type of task.

---

> > ### Author Response · Authors · 2024-08-13
> > **Part 2**
> >
> > - Search and Rescue (SAR) tasks: The rationale behind using the SAR environment was to showcase the effectiveness of our approach in scenarios where cooperation is strictly necessary (e.g. 2 agents are needed to carry a human), the success is sensitive to the time-optimality of the actions (they need to split the agents efficiently between the fires to avoid uncontrolled spread), and exploration is necessary to find the missing human. It is important to note that we propose the SAR environment as an additional benchmark to evaluate the flexibility of LLAMAR under these different constraints rather than an extraneous component of the MAP-THOR dataset. Our current environment serves as an abstraction of the core search and rescue challenges, but we can imagine a more extensive & realistic simulation so we do not propose to use it as a definitive benchmark for future experiments.
> >
> > 3. Thanks a lot for pointing out the Ok-robot paper! We were initially thinking of including a histogram of different failure cases in the revised paper. But as you pointed out, the Sankey diagram (as in Fig 4 of the Ok-robot paper) is more informative to guide future improvements. We will include this in our revised paper.

---

### Official Review · Reviewer_Eixs · 2024-07-12

**Soundness:** 4
**Presentation:** 4
**Contribution:** 4
**Rating:** 8
**Confidence:** 4

**Summary:**

This paper explores the use of Language Models (LMs) to generate task plans for autonomous robots, emphasizing adaptability across diverse tasks without the need for extensive customization. The authors introduce LLaMAR, an LM-based agent framework that employs a “plan-act-correct-verify” strategy to manage long-horizon tasks in partially observable environments. The effectiveness of LLaMAR is assessed using MAP-THOR, a new benchmark featuring multi-agent scenarios and various household tasks at different complexity levels within the AI2-THOR environment. LLaMAR demonstrates a higher success rate than competing models in these tests.

**Strengths:**

1. The paper is well-written and structured, making technical details accessible and easy to follow.
2. LLaMAR introduces a unique "plan-act-correct-verify" framework, integrating Language Models to offer a fresh approach for long-horizon planning in dynamic, partially observable multi-agent environments.
3. Instead of straightforwardly applying the general framework of an LM-based agent, LLaMAR proposes multiple essential components, particularly for solving planning problems in embodied AI, i.e., verifier, corrector, and action parsing. For example, to mitigate the hallucination of LLMs/VLMs, LLaMAR presents a verification module for feedback and self-reflection, which is important when interacting with practical environments.
4. The introduction of MAP-THOR as a test suite for evaluating the effectiveness of the planner across various household tasks underscores the robustness and practical applicability of LLaMAR, providing a sense of confidence in its potential use.

**Weaknesses:**

1. The method’s reliance on multiple LM-based modules for planning and execution might introduce significant computational overhead and complexity.
2. The detailed, step-by-step reasoning process, while beneficial for accuracy and responsiveness, could incur high costs, especially when utilizing commercial LM APIs.

**Questions:**

1. The use of fine-tuned sentence-BERT for action parsing is innovative, but the paper should discuss potential performance issues and the implications of incorrect mappings more quantitatively, for example, by comparing with more general function calling in LLMs.
2. As indicated in Algorithm 2, the corrector works after the robot performs the actions, which may cause safety problems in real-world experiments. For example, the place action is dangerous when the fridge is not open. Is it possible to pre-check the action before the actual execution? Moreover, solely relying on VLM as a verifier may be unreliable; more traditional perception tools (e.g., segmentation and detection) may help improve the system's reliability.

**Limitations:**

YES

---

> ### Author Rebuttal · Authors · 2024-08-06
>
> Dear Reviewer,
>
> Thank you for your comprehensive review and insightful feedback on our paper. We appreciate your thoughtful comments and suggestions, and we have provided our responses and clarifications below.
>
> ### SentenceBERT
>
> Yes, we agree with the concern about potential performance issues and incorrect mappings with SentenceBERT. Based on our experimental results, 96.7% of time the free-form natural language was correctly mapped to the feasible actions. The 2 main reasons for failures in the few cases it failed were:
>
> 1. Incorrect mapping of objects: when the agents had not yet explored the environment enough and the Actor module suggested the agents interact with objects that were not yet available in the agents’ combined memory. Example: The free-form output “navigate to the table” was mapped to the action “NavigateTo(ArmChair)”.
> 2. Wrong object numeration: When there are multiple objects of the same type in the memory, the sentenceBERT maps “Cabinet_3” to “Cabinet_1” which sometimes leads to incorrect actions being executed for the completion of the plan.
>
> We will add more specific examples of the failure cases of the sentenceBERT module in the appendix of our revised manuscript.
>
> As for using more general function calling with LLMs, we saw in our initial experiments that the LLMs performed well with function calling only when we gave a few examples of the exact format (eg., JSON, pythonic plans, dictionary, etc.) we wanted the LLM to answer in. This meant that we would have to use a significant amount of prompt tokens to include the examples in the input prompt. A way to get around this is to fine-tune the underlying LLMs on the type of outputs we want which we would like to do in our future work along with having specialized smaller LLMs for each module.
>
> ### Checking the safety of actions before execution
>
> This is a great point! One possible way to tackle this is to let the LMs build a world model of the environment as done in a very recent work [1,2]. This would allow the agents to reason about what would happen if certain actions were executed in the current context of the environment. This could act as a pre-execution action checker and allow for filtering unsafe or infeasible actions. However, a drawback of using another LM for the world model would be the added computational complexity.
>
> [1]: Xiang, J., Liu, G., Gu, Y., Gao, Q., Ning, Y., Zha, Y., ... & Hu, Z. (2024). Pandora: Towards General World Model with Natural Language Actions and Video States. *arXiv preprint arXiv:2406.09455*.
> [2]: Zhang, H., Wang, Z., Lyu, Q., Zhang, Z., Chen, S., Shu, T., ... & Gan, C. (2024). COMBO: Compositional World Models for Embodied Multi-Agent Cooperation. arXiv preprint arXiv:2404.10775.

---

> > ### Comment · Reviewer_Eixs · 2024-08-13
> >
> > Thank you for the comprehensive response. Your rebuttal has addressed my concerns, so I will maintain my original positive rating.

---

> > > ### Author Response · Authors · 2024-08-13
> > >
> > > Dear Reviewer Eixs,
> > >
> > > Thanks a lot for acknowledging the rebuttal. We appreciate your time and efforts in reviewing our paper. Your feedback has been valuable and we will incorporate your suggestions in the final paper.

---

### Official Review · Reviewer_nYvP · 2024-07-15

**Soundness:** 3
**Presentation:** 2
**Contribution:** 2
**Rating:** 5
**Confidence:** 4

**Summary:**

The paper addresses the problem of long-horizon household tasks in a multi-agent setup using LLMs and for this, the authors propose LLaMAR. This framework integrates planning, acting, correcting, and verifying parts. LLaMAR plans a high-level action sequence, assigns them to multiple agents, and checks if actions fail to be executed, followed by a heuristic exploration strategy. It also converts the planned actions to the admissible ones in a retrieval manner based on text similarity. The approach is validated in MAP-THOR, an extended version of AI2-THOR that includes household tasks in the multi-agent setting.

**Strengths:**

- The paper tackles an important problem of LLM-based multi-agent scenarios.
- The proposed pipeline is straightforward and easy to replace due to its modular nature, yet raising concern about its justification (see weaknesses).
- The paper tries to remove unrealistic aspects of prior work, such as no perfect execution of low-level primitives, which potentially improves the deployability of the proposed approach to real-world scenarios.

**Weaknesses:**

- The proposed framework takes a plan-act-correct-verify pipeline but this is not well-justified. Why do we need the four components in that order? As each plan, act, correct, and verify part has been actively investigated in the literature, the core contribution may come from how we combine them.
- L49 says that the proposed approach does not rely on oracle feedback, but L176 says it uses the simulator to check whether an action is executed successfully, which is contradictive. Using such GT action failure checking seems quite unrealistic.
- One argued contribution is MAP-THOR, but the current description (L207-L219) lacks the details of it. How many (training/evaluation) episodes are used? How different is MAP-THOR compared to previous benchmarks for various aspects? How to collect the dataset? Summarizing them in some tables with qualitative analysis might be helpful.
- What are the values in the parentheses in the tables? Describing what they are in the table captions might improve readability. Are they min/max values for the corresponding metrics?
  - If yes, then the variation seems quite high. For example, in Table 3, the min of "LLaMAR (w/ exploration)" is lower than the min of "LLaMAR (w/o exploration)," which raises a concern about the outperformance argued.

**Questions:**

See weaknesses above.

**Limitations:**

The authors do not address the potential negative societal impact.

---

> ### Author Rebuttal · Authors · 2024-08-06
>
> Dear Reviewer,
>
> Thank you for your comprehensive review and insightful feedback on our paper. We appreciate your thoughtful comments and suggestions, and we have provided our responses and clarifications below.
>
> ### Justification for the Proposed Framework
>
> The specific order of the modules is due to natural causal relationships in which environment feedback is received, which restricts our freedom in permuting the order significantly.
>
> We use the planner as the first module because it allows LLaMAR to come up with an initial list of open subtasks that could be completed based on the current observation and past memory to satisfy the task. This list serves as a rough high-level plan. The actor then uses this information to suggest the necessary actions.
>
> The Corrector is used after the Actor module to identify reasons for failures in the execution of the actions suggested by the Actor. Note that the failure module is inert and only suggests corrective actions. Only the Actor module decides the final actions to be executed. This role distinction allows for clear reasoning on failures when they occur and lets the actor module focus on choosing actions.
>
> The verifier is used after the action is executed to update the list of closed subtasks so that LLaMAR can be current with the progress toward the completion of the environment task. This allows the planner to update the list of open subtasks in the next step. In essence, the planner and the verifier ensure that the progress of the agents is tracked and the actor and the corrector ensure that the actions are executed successfully to advance towards completion of the task.
>
> Our revised manuscript will include this justification with illustrative examples so as to explain why we converged on this solution.
>
> ### Oracle feedback clarification
> The terminology is confusing and should be cleared up. Thanks a lot for pointing it out. In L49, when we say our approach does not rely on oracle feedback, we mean to say that the *environment* does not provide any feedback. Instead, it uses controller feedback—a boolean variable indicating action success or failure, such as obstacles in planned paths or failed object grabs.
>
> It is indeed true that, in general, loss of that low-level controller information could make the problem harder, and it is a fruitful direction to explore in future work.
>
> ### MAP-THOR details
>
> MAP-THOR includes 45 evaluation scenarios each initialized in 5 different rooms in AI2THOR. All of these 45 tasks are described in Appendix C. We refer to L726-L787 for a detailed list of all the tasks in MAP-THOR. A more detailed description of how to use the dataset to run experiments will be provided in the code’s repository when it is open-sourced. Additionally, a description of how to append new tasks to MAP-THOR will also be provided there. MAP-THOR as a dataset consists of AI2Thor scene pre-initializations, task information (task description, timeout, etc.) pairs, and an oracle performance evaluator (to get the metrics). Thus, there isn’t an explicit collection procedure since we create and simulate the tasks based on each floor plan.
>
> The MAP-THOR dataset was not used to fine-tune/ train any of the modules in the cognitive architecture. Indeed we should provide a fixed training and evaluation division of the dataset in order to aid future work.
>
> MAP-THOR is designed to include tasks that could be done by multiple agents. Previous benchmarks like AlfWorld, and SmartLLM include many tasks that could be done with just a single agent. Example: “Slice the tomato”, “throw the spatula in the trash”, etc. Link:https://github.com/SMARTlab-Purdue/SMART-LLM/blob/master/data/final_test/FloorPlan6.json. Although many of the tasks in these benchmarks require single agents, there are a few tasks that could be solved with multiple agents. We include those tasks in our benchmark. Other benchmarks like the ones used in CoELA in the ThreeDWorld and CWAH environments include tasks that require multiple agents. But there are only limited tasks, 5-10, in those experiments. Another major difference in MAP-THOR is that we instantiate each task in 5 different rooms with random initialization to test for the robustness of the underlying LM based cognitive architecture to variations in the room layouts. This allows us to perform a systematic study of its performance when compared to previous approaches. Along with this, we have created a script to add custom scenarios in AI2THOR. We will add more details about its usage in our documentation of the code and in our appendix along with pointing out specific details about how it is qualitatively different than previous approaches.
>
> ### Clarification on the metrics in tables
>
> The metrics shown in all tables are means. The values in the parentheses are the 95% confidence intervals. Since success rates are binomial, we use the Clopper-Pearson Interval as the confidence interval. For the metrics: “Success Rates (SR)”, “Transport Rates (TR)”, “Coverage (C)” and “Balance(B)”, higher is better and for “Steps (L)” lower is better
>
> In Table 3, the means for SR, TR, C, and B are higher for LLaMAR (w/ exploration) compared to LLaMAR (wo/ exploration). Similarly, the lower bound of the confidence intervals for SR, TR, and C are higher for LLaMAR (w/ exploration). Similarly, the upper bounds for SR, TR, C, and B for LLaMAR (w/ exploration) are higher than or equal to the upper bounds for LLaMAR (wo/ exploration). For Balance (B), the means are equal for w/ and w/o exploration and the lower bound for w/ exploration (0.75) is slightly smaller than the lower bound for w/o exploration (0.77)

---

> > ### Comment · Reviewer_nYvP · 2024-08-11
> > **Response by Reviewer nYvP**
> >
> > Thank you for your detailed response. The rebuttal addresses my concerns and thus I am happy to increase my score.

---

> > > ### Author Response · Authors · 2024-08-12
> > >
> > > Thank you for acknowledging the rebuttal and increasing the score. We appreciate your time and efforts in reviewing our paper. We will incorporate your suggestions into the final paper.

---

### Author Rebuttal · Authors · 2024-08-06

Dear reviewers,

We would like to thank you all for your comprehensive reviews and insightful feedback on our paper. We appreciate your thoughtful comments and suggestions. A few of the reviewers pointed out that some of the tasks do not require explicit cooperation within the multi-agent team and the performance metrics degraded when the number of agents was increased from 3 to 5. The decrease in metrics can be attributed to the rooms in MAP-THOR becoming crowded with 4 and 5 agents hence blocking the agents from navigating without colliding with other agents. We wanted to address this with an experiment with our proposed approach LLaMAR in a different environment.

We implemented a partially observable search & rescue and fire relief environment for multi-agent systems in a grid world. Depending on the scene, there is a mix of missing people to be found, and wildfires to be stopped before they spread geographically.

- Each fire is composed of a large flammable region with a fixed set of sources that spread through time. The higher the intensity, the faster the fire will spread. The fires can be of class A or B, extinguished through the use of water and sand respectively. Both these resources can be collected through resource reservoirs spread geographically.
- Each person is initially stranded in an unknown location. The goal is to rescue and transport each person to a drop-off location (known apriori). The person must have two agents simultaneously carrying them to be transported to the drop-off location.

We provide a justification to why this task is relevant to long-term planning and multi-agent collaboration below.

### *Multi-agent nature*

- Scalability - prevents crowdedness, can increase complexity
    - With the addition of multiple parallel emergency disaster scenarios at once, the environment scales comfortably to more agents. Additionally, the complexity slowly increases as the model has to coordinate between more actors and competing emergencies.
- Task require explicit cooperation
    - In order to move a group of humans, at least 2 agents are required. Thus, explicit multi-agent collaboration is required.
    - Due to the time pressure, in order to successfully stop the fire, drones must all effectively collaborate in collecting and strategically using resources in high intensity regions.
- Exploration task assignment
    - There is an explicit tradeoff between allocating drones towards exploring to find the lost people, and fighting the current fires.

### *Long-term planning*

- Dependency chain
    - In order to resolve a fire it requires that the source is identified, the type of fire is identified, and the appropriate resources to stop it are acquired and then used.
- Uncertainty
    - Inherent uncertainty as to where the lost people can be found, and at what point in the timeline they will.
- Irreversibility and forced consequences
    - With a fire that spreads, present actions have irreversible future consequences. A balance needs to be struck between fighting the fire’s source (to stop it from continuing) versus a periphery (to prevent geographic spread).

### Description of all scenes:

Each scene has one reservoir for each fire type.

- *Scene 1* - (one) Type A fire with 2 initial sources, (one) Type B fire with 1 initial source, and (one) lost person
- *Scene 2* - (one) Type A fire with 1 initial source, (one) Type B fire with 2 initial sources, and (one) lost person
- *Scene 3* - (two) Type A fires each with 1 initial source, (one) Type B fires with 1 initial source, and (one) lost person
- *Scene 4* - (one) Type A fire each with 3 initial sources, and (one) lost person
- *Scene 5* - (one) Type A fire with 1 initial source, (one) Type B fire with 1 initial source, and (two) lost people

We evaluate each scene on 5 different random seeds.

## Results

The metrics shown are means along with the 95% confidence intervals shown in parentheses. Since success rates are binomial, we use the Clopper-Pearson Interval as the confidence interval. For the metrics: “Success Rates (SR)”, “Transport Rates (TR)”, “Coverage (C)” and “Balance(B)”, higher is better and for “Steps (L)” lower is better

| # agents | Success Rate | Transport Rate | Coverage | Balance | Steps |
| --- | --- | --- | --- | --- | --- |
| 2 | 0.44 (0.24, 0.65) | 0.86 (0.79, 0.94) | 0.94 (0.88, 0.99) | 0.91 (0.88, 0.95) | 27.76 (24.15, 30) |
| 3 | 0.68 (0.46, 0.85) | 0.92 (0.86, 0.98) | 0.96 (0.91, 1.0) | 0.80 (0.73, 0.86) | 21.88 (17.83, 25.92) |
| 4 | 0.72 (0.50, 0.85) | 0.94 (0.88, 0.98) | 0.98 (0.93,1.0) | 0.78 (0.74, 0.83) | 22.00 (17.96, 26.03) |
| 5 | 0.74 (0.52, 0.86) | 0.96 (0.94, 0.99) | 1.0 (1.0, 1.0) | 0.73 (0.67, 0.79) | 24.52 (20.24, 28.79) |

## Analysis of results

There is an observed increasing trend in success rate, transport rate, and coverage when the number of agents increases from 2 to 5 suggesting that more agents lead to better completion of the tasks on average.

The balance metric, however, follows a monotonically decreasing trend from 2 to 5 agents. This would be expected due to the increasing hardness of equal task allocation between many agents. Similarly, the steps metric shows a decrease from 2 to 3, but monotonic increases from 3 through 5. This fact can similarly be expected due to emergent inefficiencies in the performance of many agents.

We will include these in our revised manuscript.

---

### Decision · Program_Chairs · 2024-09-25

**Decision:**

Accept (poster)

**Comment:**

This paper presents LM-based Long-Horizon Planner for Multi-Agent Robotics (LLaMAR) an approach that combines LLMs in a  plan-act-correct-verify framework to solve long-horizon multi-agent tasks. It also introduces the MAP-THOR test suite for evaluating results. The proposed method performs well in the experiments.

LLMs are becoming more widely used but current methods are still not very good at complex, partially observable, long-horizon tasks. Developing methods that can solve more challenging tasks is an important area and the proposed method is a promising step in this direction.

There are some questions about the motivation and details of the particular architecture, the high computational complexity, and whether the current approach is really multi-agent but the authors addressed them in the rebuttal/discussion. The authors should include these points (as well as other feedback from the reviewers) in the final paper.